https://doi.org/10.1038/s42003-023-05339-3　　OPEN

# W546 stacking disruption traps the human porphyrin transporter ABCB6 in an outward-facing transient state

Sang Soo Lee[1], Jun Gyou Park[1], Eunhong Jang[1], Seung Hun Choi[1], Subin Kim[1], Ji Won Kim[2] & Mi Sun Jin [1✉]

Human ATP-binding cassette transporter subfamily B6 (ABCB6) is a mitochondrial ATP-driven pump that translocates porphyrins from the cytoplasm into mitochondria for heme biosynthesis. Within the transport pathway, a conserved aromatic residue W546 located in each monomer plays a pivotal role in stabilizing the occluded conformation via π-stacking interactions. Herein, we employed cryo-electron microscopy to investigate the structural consequences of a single W546A mutation in ABCB6, both in detergent micelles and nanodiscs. The results demonstrate that the W546A mutation alters the conformational dynamics of detergent-purified ABCB6, leading to entrapment of the transporter in an outward-facing transient state. However, in the nanodisc system, we observed a direct interaction between the transporter and a phospholipid molecule that compensates for the absence of the W546 residue, thereby facilitating the normal conformational transition of the transporter toward the occluded state following ATP hydrolysis. The findings also reveal that adoption of the outward-facing conformation causes charge repulsion between ABCB6 and the bound substrate, and rearrangement of key interacting residues at the substrate-binding site. Consequently, the affinity for the substrate is significantly reduced, facilitating its release from the transporter.

[1] School of Life Sciences, GIST, 123 Cheomdan-gwagiro, Buk-gu, Gwangju 61005, Republic of Korea. [2] Department of Life Sciences, POSTECH, 77 Cheongam-Ro, Nam-gu, Pohang 37673, Republic of Korea. ✉email: misunjin@gist.ac.kr

Human ATP-binding cassette transporter subfamily B6 (ABCB6) is a mitochondrial porphyrin transporter that uses energy from ATP hydrolysis to translocate the heme precursors protoporphyrin IX and coproporphyrinogen III from the cytoplasm into mitochondria for heme biosynthesis[1]. ABCB6 is also highly expressed in the Golgi apparatus[2], lysosomes[3], plasma membrane[4], and intracellular vesicles[5]. Therefore, ABCB6 may perform diverse roles beyond mitochondrial porphyrin transport[6], potentially including protecting cells against oxidative stress[7], promoting heavy metal tolerance[8,9], and stimulating broad resistance to xenobiotics[10–15]. In addition, ABCB6 serves as a carrier for the Langereis (Lan) antigen, and its various mutation types define the Lan blood group system, which is characterized by the absence of Lan antigen on red blood cells[16]. Genetic mutations of ABCB6 are associated with many tissue-specific diseases including porphyria, dyschromatosis universalis hereditaria (DUH)[17], ocular coloboma[18], pseudohyperkalemia[19], and atherosclerosis[20].

ABCB6 is a Type IV half-transporter of 842 amino acids with only one transmembrane domain (TMD) and one cytosolic nucleotide-binding domain (NBD) with an extra N-terminal TMD (TMD0)[21]. It functions as a homodimer with the substrate transport pathway and the active site of ATP hydrolysis built from residues from both chains. TMD0 is a unique domain that folds independently of the rest of the protein chain. It is crucial for endo-lysosomal targeting of ABCB6 but dispensable for ATPase activity, substrate transport, and physiological dimerization[22]. A high-resolution structure of isolated NBD from human ABCB6 was determined by X-ray crystallography, revealing architectural similarity to other known NBD structures[23]. Efforts to obtain high-resolution structures of full-length human ABCB6 using cryo-electron microscopy (cryo-EM) have encountered challenges, including limited resolution (5.2 Å) in the apo state due to inherent flexibility and conformational heterogeneity of the TMD0 domain[24]. Moreover, during the expression and purification of full-length protein in insect cells, partial proteolysis was observed, adding another layer of complexity to structural analysis[25]. Subsequent investigations successfully addressed these challenges by focusing on the core region without TMD0 (hereafter referred to as hABCB6core), leading to elucidation of near-atomic-resolution structures of the transporter in various conformational states, including apo inward-facing, substrate-bound inward-facing, and ATP-bound occluded forms[24–26]. While these advances are significant, the complete conformational cycle of ABCB6 remains uncertain, primarily due to the lack of structural characterization for the outward-facing transient state.

Here, we present cryo-EM structures of ADP·VO4-bound hABCB6core using both detergent and nanodisc systems. Our findings provide valuable insight into the role of the W546A mutation and protein-lipid interactions in modulating the conformational preferences and dynamics of ABCB6. By integrating our results with previous data, we construct a more complete model of the conformational changes underpinning the substrate release and catalytic cycle of ABCB6.

## Results

**$VO_4^{3-}$ and $AlF_4^-$ inhibit the catalytic cycle of ABCB6core, but $SO_4^{2-}$ does not.** Many ATP-binding cassette (ABC) transporters can be captured in an outward-facing or occluded conformation in the presence of nucleotide that mimics different stages of the ATPase cycle[27–30]. Song et al. (2021) reported that when ATP binds alone without a substrate, the catalytically inactive E752Q mutant of hABCB6core adopts a predominantly occluded conformation in which the NBDs form a closed dimer and the substrate-binding site is no longer accessible from either side of the membrane[26]. This knowledge allowed us to hypothesize that such an ATP-bound, pre-hydrolytic state effectively traps hABCB6core in the occluded configuration (pre-occluded). Furthermore, this hypothesis led us to postulate that other hydrolytic stages of the ATPase cycle, such as the post-hydrolysis state or the transition state leading to it, may potentially trap the transporter in an outward-facing form[31,32]. Through NADH-coupled ATPase assays[33], we confirmed that the ATPase activity of hABCB6core is strongly suppressed by the presence of $VO_4^{3-}$ or $AlF_4^-$, comparable to observations for the E752Q mutant (Fig. 1a and Supplementary Fig. 1). Consistently, these transition state analogs also block the transport activity of hABCB6core towards the well-known substrate coproporphyrin III (CPIII; Fig. 1b and Supplementary Fig. 2). By comparison, the $SO_4^{2-}$ ion does not completely prevent hydrolysis (Fig. 1a). This indicates that the sulfate

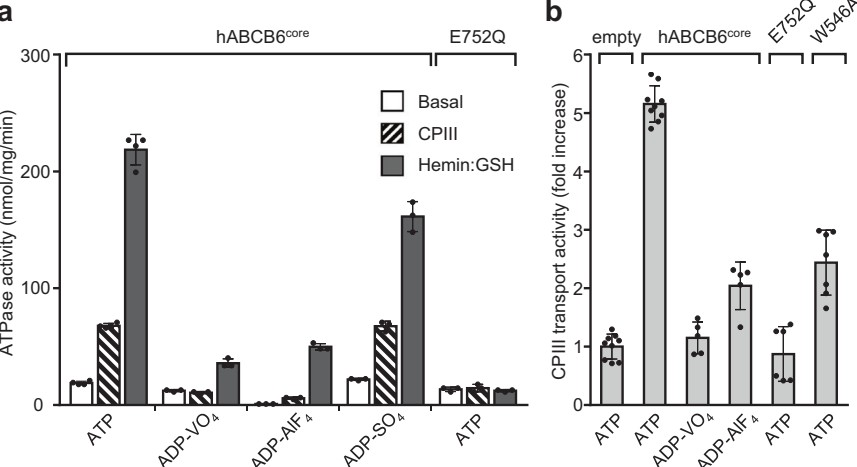

**Fig. 1 Inhibition of the functional activity of hABCB6core by ATP analogs. a** ATPase activities of detergent-purified hABCB6core and the E752Q mutant with and without porphyrin substrates (20 μM CPIII or 10 μM hemin: 1 mM GSH) in the presence of 10 mM $Mg^{2+}$ and 2 mM ATP plus $VO_4^{3-}$, $AlF_4^-$, or $SO_4^{2-}$. Results are means ± standard deviation (SD) of more than three independent experiments. **b** Inhibition of the CPIII transport activity of hABCB6core (or its variants) by different ATP analogs. Liposome-based transport activity was measured in the presence of 50 μM CPIII, 10 mM $Mg^{2+}$, and 4 mM ATP (or its derivatives). The activity of protein-free liposomes was set at 100%. Results are means ± SD of at least five independent measurements from two different liposome preparations.

group is not effective in stabilizing the transporter in a specific conformation by displacing the ATP hydrolysis product Pi. Considering the extent of activity inhibition, we selected ADP·VO$_4$ and employed a single-particle cryo-EM approach to examine whether hABCB6$^{core}$ can indeed be trapped in its outward-facing state during the post-hydrolysis transition phase.

**A potential strategy to promote the outward-facing conformation in hABCB6$^{core}$.** For cryo-EM analysis of the outward-facing conformation of hABCB6$^{core}$, detergent-purified protein was incubated with 100 μM CPIII, 2 mM Mg$^{2+}$/ATP, and 2 mM sodium orthovanadate at 22 °C for 30 min before grid preparation. A cryo-EM dataset consisting of 10,280 movies was collected and preliminarily analyzed using RELION[34] and CryoSPARC[35] to evaluate the conformational states of hABCB6$^{core}$ under those experimental conditions. As expected, the outward-facing conformation was evident in the presence of ADP·VO$_4$ and the ATPase-stimulating substrate CPIII (79,317 particles; Fig. 2a, b, and Supplementary Fig. 3). However, a large number of particles still adopted inward-facing (210,460 particles) or post-translocation, occluded (137,025 particles, post-occluded) conformations. This observation may seem contradictory to functional data indicating that the ATPase and transport activities of hABCB6$^{core}$ are fully inhibited in the presence of vanadate (Fig. 1a, b). We propose two plausible explanations for this contradiction: firstly, it is possible that ATPase- or transport-incompetent molecules persist in the inward-facing conformation, even when vanadate is present; secondly, the exceptional stability of the inward-facing or post-occluded conformation of hABCB6$^{core}$ hinders its transition to or entrapment in the outward-facing state. Based on these speculations, we predicted that engineering hABCB6$^{core}$ to enhance its functional competence or disrupt its inherent conformational stability in the inward-facing or post-occluded state could be a viable strategy to promote the formation of the desired outward-facing conformation.

With this rationale, we looked more closely at the hABCB6$^{core}$ structures in the inward-facing (Fig. 2b, class 1) and post-occluded (Fig. 2b, class 2) states. This analysis led us to specifically examine residue W546, located in the two pairs of TM11 helices near the 2-fold axis (Fig. 2c). Previously, this highly conserved aromatic residue (Fig. 2d) was shown to play a crucial role in capping the binding site, effectively sequestering substrates within the central cavity[25,26]. Despite a slight alteration in the orientation of W546 (Fig. 2e), structural comparison indicates that pre- and post-occluded structures are almost identical, with a Cα r.m.s.d. of 0.9 Å (Supplementary Fig. 4). Considering the positioning of W546 along the substrate transport pathway, it is reasonable to hypothesize that this contact would be disrupted during formation of the outward-facing state. Hence, we predicted that deletion of W546 could potentially disrupt the conformational balance of hABCB6$^{core}$ and promote a shift towards the outward-facing state.

**Functional effects of W546A mutation on ATPase and transport activities of hABCB6$^{core}$.** To confirm the proposed role of W546 as a structural stabilizer in the occluded state, we compared the functional properties of the W546A mutant generated in previous work with those of hABCB6$^{core}$ [25,26]. Detergent-purified hABCB6$^{core}$ exhibited basal ATPase activity with a V$_{max}$ of 18.3 nmol/mg/min (Fig. 3 and Supplementary Table 1). The ATPase activity of hABCB6$^{core}$ reconstituted in nanodiscs was 5.2-fold higher (96.6 nmol/mg/min) than that of the hABCB6$^{core}$ in detergent micelles. By contrast, the W546A mutant exhibited a comparable ATPase turnover rate regardless of the presence or absence of lipids, and regardless of the lipid composition of the

nanodiscs. Consequently, the ATPase activity of the W546A mutant was 2.7-fold higher (48.9 nmol/mg/min) than that of the hABCB6$^{core}$ in detergent micelles, while its activity in nanodiscs, composed of porcine brain polar lipids, was 2.4-fold lower (39.6 nmol/mg/min). Irrespective of the conditions, the K$_m$ remained relatively unaffected (57 ~ 73 μM). These kinetic values differ somewhat from those previously reported. For example, Wang et al. (2020) demonstrated that both the full-length human ABCB6 and its W546A mutant have basal ATPase activity with a V$_{max}$ of ~40 nmol/mg/min in LMNG (lauryl maltose neopentyl glycol) micelles[24]. Song et al. (2021) showed that hABCB6$^{core}$ has negligible basal ATPase activity in C12E9 (dodecyl nonaethylene glycol ether) micelles[26]. However, the protein regained basal ATPase activity with a V$_{max}$ ranging from 27 to 32 nmol/mg/min upon reconstitution in nanodiscs composed of POPC (1-Palmitoyl-2-oleoyl-sn-glycero-3-phosphocholine) and POPE (1-palmitoyl-2-oleoyl-sn-glycero-3-phosphoethanolamine) at a ratio of 3:1 (w/w). The ATPase activity of the full-length protein reconstituted in liposomes exhibited a much higher value, reaching a V$_{max}$ of 492.3 nmol/mg/min[36]. It is speculated that this discrepancy in the kinetic data for ABCB6 across different research groups may arise from multiple factors, including differences in protein purification conditions and lipid composition of the nanodiscs, the presence or absence of TMD0 in the construct, and utilization of different assay methods. Nevertheless, in the detergent solution, the substantial increase in the basal turnover of the W546A mutant compared to hABCB6$^{core}$ strongly suggests that disruption of the W546-W546' stacking interaction reduces the intrinsic stability of the occluded state. As a consequence, the W546A mutation is believed to increase the tendency of the transporter to adopt an outward-facing conformation. Interestingly, contrary to our initial expectation based on the observed increase in ATPase activity for the W546A mutant (Fig. 3), transport assays revealed the opposite effect. Specifically, the mutant exhibited ~2-fold lower CPIII transport activity than the hABCB6$^{core}$ protein (Fig. 1b and Supplementary Fig. 2). This unexpected result can be reconciled by the fact that W546 is a key residue involved in porphyrin binding[25]. Therefore, the tryptophan-to-alanine mutation likely impairs CPIII transport activity by decreasing the binding affinity.

**The W546A mutant favors the outward-facing state in detergent but not in nanodiscs.** To evaluate the structural impact of W546A mutation, we prepared grids as mentioned above using detergent-purified or nanodisc-reconstituted W546A mutant (Supplementary Fig. 5). As expected, the first 3D classification analysis revealed that most of the detergent-purified mutant molecules (59.5%) adopted an outward-facing conformation (Fig. 4a, b, and Supplementary Figs. 6−8). However, a significant proportion of molecules (36%) still adopted the post-occluded conformation. The occluded structures reveal that, besides W546 on TM 11, residues M542, Y550, and M553 also play crucial roles in establishing interactions between the two halves of the transporter within a 5 Å distance (Fig. 2e)[26]. These additional interactions may contribute to maintaining hABCB6$^{core}$ occlusion by causing another stabilizing effect, even in the absence of W546.

In contrast to the W546A mutant in detergent micelles, the nanodisc-reconstituted protein predominantly exists in the inward-facing or post-occluded states (Fig. 4c, d, and Supplementary Figs. 9−11). The difference in the conformations of the W546A mutant in detergent solution and lipid environment may reflect the inherent complexity and adaptability of ABCB6 to its surroundings. This adaptability may be further influenced by direct interactions between the transporter and membrane lipids[37]. Consistent with this hypothesis, we observed the electron

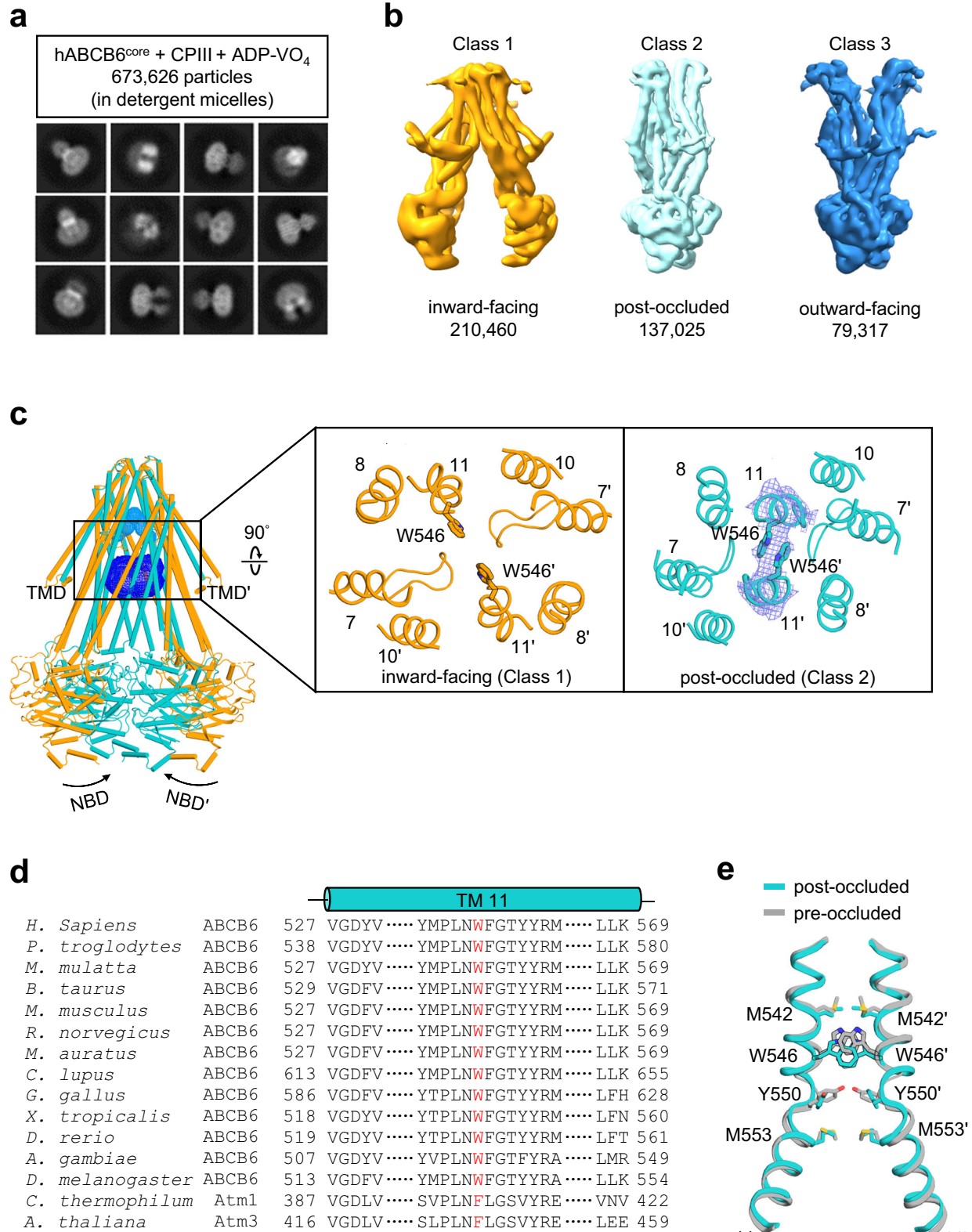

densities corresponding to ordered lipids in the annular belt near the hydrophobic surface of transporters within nanodiscs (Supplementary Fig. 12). Notably, in the post-occluded state, we observed strong electron density within a hydrophobic groove formed by the elbow helix, TM 9, and TM 11 of each monomer (Fig. 5a). The density shape corresponds to a phospholipid molecule with two fatty acyl chains. The nanodiscs used in our structural studies were composed of major brain polar lipids derived from porcine sources, including phosphatidylethanolamine (PE, 33.1%), phosphatidylserine (PS, 18.5%), and phosphatidylcholine (PC, 12.6%). Therefore, we constructed an atomic model of the POPE (or 16:0-18:1 PE) molecule in the head-down orientation as a representative phospholipid model (Fig. 5b, c). Our analysis revealed that POPE interacts with highly conserved

**Fig. 2 The occluded state of hABCB6$^{core}$ is highly stabilized by W546-W546' interaction. a**, **b** Representative reference-free 2D class averages (**a**) and 3D classification (**b**) of hABCB6$^{core}$ in the presence of CPIII and Mg$^{2+}$/ADP·VO$_4$. The bottom numbers in (**b**) refer to the number of particles used to compute each 3D class. See also Supplementary Fig. 3. **c** Structural comparison of hABCB6$^{core}$ in inward-facing (Class 1 from Fig. 2b) and occluded (Class 2 from Fig. 2b) states viewed from the plane of the membrane (left) and the extracellular side (right). The upper and central cavities are drawn as cyan and blue mesh, respectively. Mg$^{2+}$/ ADP·VO$_4$ molecules in the occluded structure are omitted for simplicity. The EM density (blue mesh) of the W546 residue is contoured at the 4 σ level. Note that the residues (or helices) of one monomer are numbered with single primes to differentiate them from those of the other monomer. **d** Sequence alignment of TM 11 of ABCB6 homologs. Conserved tryptophan residues (or phenylalanine residues in bacterial orthologs) are highlighted in red. **e** Residues of TM 11 involved in stabilization of the pre- (PDB ID 7EKL) and post-occluded conformations are shown as sticks.

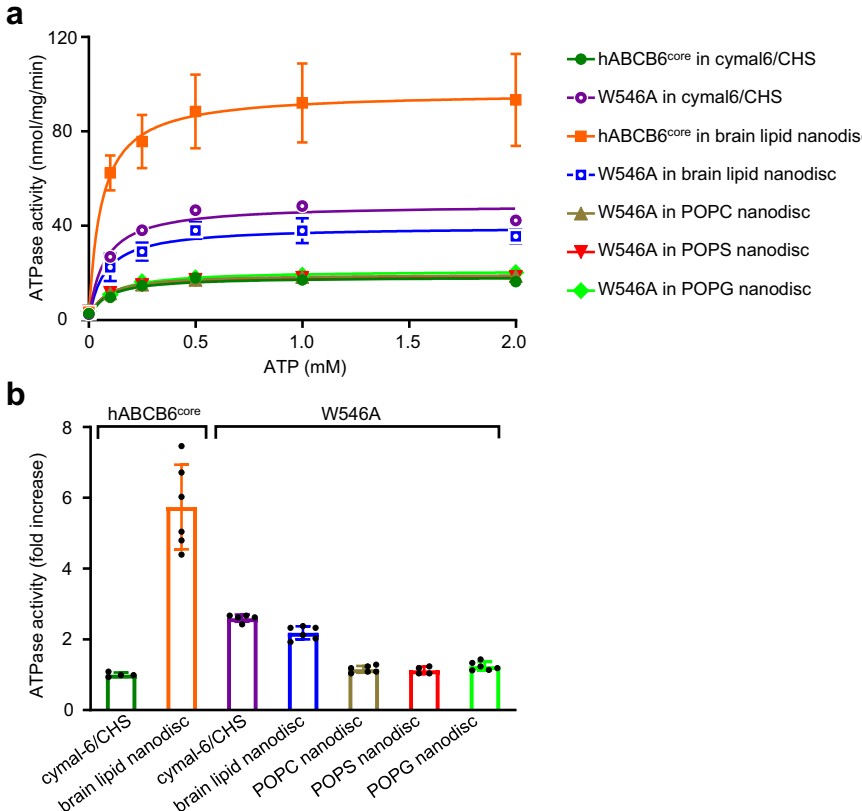

**Fig. 3 ATPase activity of hABCB6$^{core}$ and W546A proteins in detergent micelles or various nanodiscs. a** Comparison of the ATPase activities as a function of ATP concentration. Results are expressed as means ± standard deviation (SD) from at least four separate experiments using the two different protein preparations. **b** At saturating concentration (2 mM ATP), maximal basal activities are represented as fold increase. The activity of detergent-purified hABCB6$^{core}$ was set at 100% as the reference. See also Supplementary Table 1.

residues across species (Fig. 5d). Furthermore, given the presence of a similar lipid density in the pre-occluded state (Supplementary Fig. 13)[26], it is evident that a phospholipid bound to the TMD groove acts as a 'wedge' to stabilize the occlusion state of the transporter. Our analysis also reveals that in the outward-facing state, TM 9 and TM 11 of the TMD groove come into closer contact, resulting in the disappearance of the lipid-binding site (Supplementary Fig. 14). This observation suggests that the role of lipids as a structural stabilizer may not be effective in this conformation. Collectively, our findings suggest that direct protein-lipid interactions contribute to the increase in the relative stability of the inward-facing or occluded states. As a result, it seems likely that the thermodynamic threshold for the transition between the outward-facing and other functional states is elevated in the lipid environment compared with detergent micelles, thereby limiting the propensity for the transporter to adopt the outward-facing conformation.

**Overall structure of hABCB6$^{core}$-W546A in the outward-facing state**. The final EM density map of the detergent-purified W546A

mutant in the outward-facing state was obtained from the best 3D class containing 56,293 particles (Supplementary Fig. 6) at a resolution of 3.9 Å with C1 or C2 symmetry, which enabled us to build an almost complete model of the transporter (Supplementary Fig. 7). Since the final model did not include electron density corresponding to the W546 residue, its side chain was positioned by an automated homology modeling server[38]. The structure has the typical features expected for the outward-facing conformation of an ABC transporter (Fig. 6a, b and Supplementary Fig. 15a). The TMDs form a V-shaped translocation pathway that is both open to the extracellular space and laterally to the membrane outer leaflet. There was no density corresponding to CPIII, suggesting that the bound substrate is exported from the transporter prior to ATP hydrolysis and ADP/Pi release. In the outward-facing structure, the cytoplasmic side of the TMDs are in close spatial proximity, contributing to transporter stability through interactions with neighboring amino acids (Fig. 6a). However, the structure has relatively poor electron density and high B-factors for the extracellular side of TMDs, suggesting that these regions undergo highly dynamic motions to promote substrate release

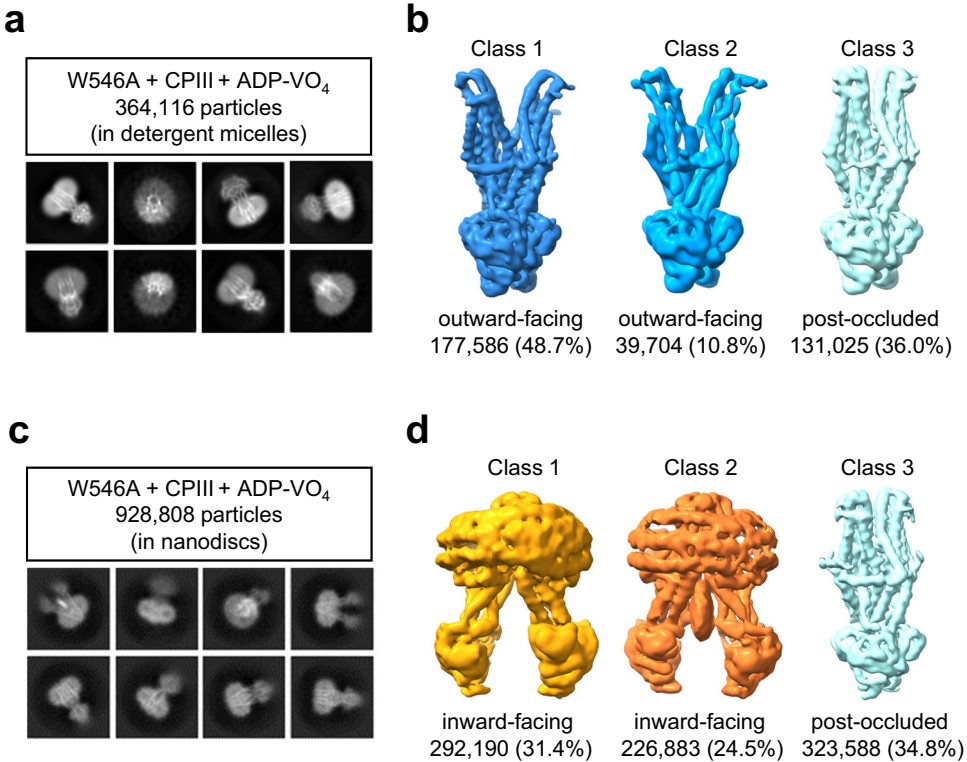

**Fig. 4 Comparison of cryo-EM maps of the hABCB6^core-W546A mutant in detergent micelles and nanodiscs. a**, **b** Cryo-EM analysis of the W546A mutant in detergent micelles, showing representative 2D class averages (**a**) and 3D classification (**b**). The bottom numbers in (**b**) refer to the number of particles used to compute each 3D class. See also Supplementary Fig. 6. **c**, **d** Cryo-EM analysis of the W546A mutant reconstituted into nanodiscs, showing selected 2D class averages (**c**) and 3D classification (**d**). The bottom numbers in (**d**) refer to the number of particles used to compute each 3D class. See also Supplementary Fig. 9.

(Supplementary Fig. 15b). Comparison of the outward-facing and occluded states revealed that the NBD structure and nucleotide-binding mode are nearly identical despite differences in the topological arrangement of the TMDs (Fig. 6c and Supplementary Fig. 16)[26]. In the NBD dimer, the two $Mg^{2+}/ADP \cdot VO_4$ molecules are sandwiched at the interface, interacting with the Walker A and B motifs, the A- and Q-loop residues of one NBD, and the ABC signature motif (LSGGE) of the opposing NBD (Fig. 6d, e). The $VO_4$ ion adopts a trigonal bipyramidal geometry that mimics the γ-phosphate undergoing nucleophilic attack by a water molecule.

**The outward-facing conformation has low substrate affinity, facilitating its release**. With the availability of new conformations of hABCB6^core in the outward-facing and post-occluded states, we can now depict the molecular features of substrate export by ABCB6. In the previous inward-facing apo structure, the cytoplasmic entrance to the central cavity is highly positively charged and thus plays an important role in attracting porphyrin molecules with negatively-charged propionic groups[24,26]. However, adoption of the outward-facing conformation exposes negatively-charged residues E346, D374, E446, and E487 at the bottom of the cavity (Fig. 6f). This negatively-charged surface appears to promote substrate release into the extracellular space by pushing porphyrins along the translocation pathway via charge repulsion, while simultaneously preventing exported substrates from re-entering the cavity.

We previously identified the residues responsible for porphyrin binding in hABCB6^core[25]. Based on this knowledge, we propose that ABCB6 employs two distinct binding mechanisms depending

on the involvement of metal ions in porphyrin molecules. Specifically, ABCB6 binds to metal-free CPIII, but it binds to a metal-centered hemin in conjunction with two molecules of glutathione (GSH; γ-glu-cys-gly). Structural comparison indicates that CPIII binding reorients the residues W546, Y550, and M553 towards the porphyrin ring (Fig. 7a, upper left and right panels)[25,26]. These movements appear to strengthen the interaction between the transporter and the bound substrate. Conversely, adoption of an outward-facing conformation shifts the side chains of key residues away from the substrate-binding site (Fig. 7a, lower right). As a result, the site appears to lose affinity for the substrate and no longer maintains its structural integrity. Together with the negatively-charged surface inside the cavity (Fig. 6f), these structural rearrangements explain why no substrate density was observed in the outward-facing conformation, even though its structure was determined in the presence of 100 μM CPIII, ~5- and 12-fold higher than the measured $K_m$ (19 ± 3.7 μM) and $K_d$ (8.2 ± 0.1 μM) values, respectively[25]. Similarly, we observed that the residues involved in GSH binding underwent rearrangement in the outward-facing state, resulting in their spatial separation from GSH and the loss of direct interaction (Fig. 7b). This, in turn, causes ABCB6 to lose its ability to bind hemin. Formation of the post-occluded state after substrate release reorients the TMDs closer together at the two-fold dimer axis such that the central cavity is no longer present (Fig. 7a, lower left)[26]. Subsequent release of ADP and Pi elicits conformational relaxation, resetting the transporter to the apo state in preparation for another catalytic cycle. However, in the absence of substrate, hABCB6^core may exhibit distinct conformational switching behavior. One possibility is that the transporter may preferentially transition from the inward-facing resting state

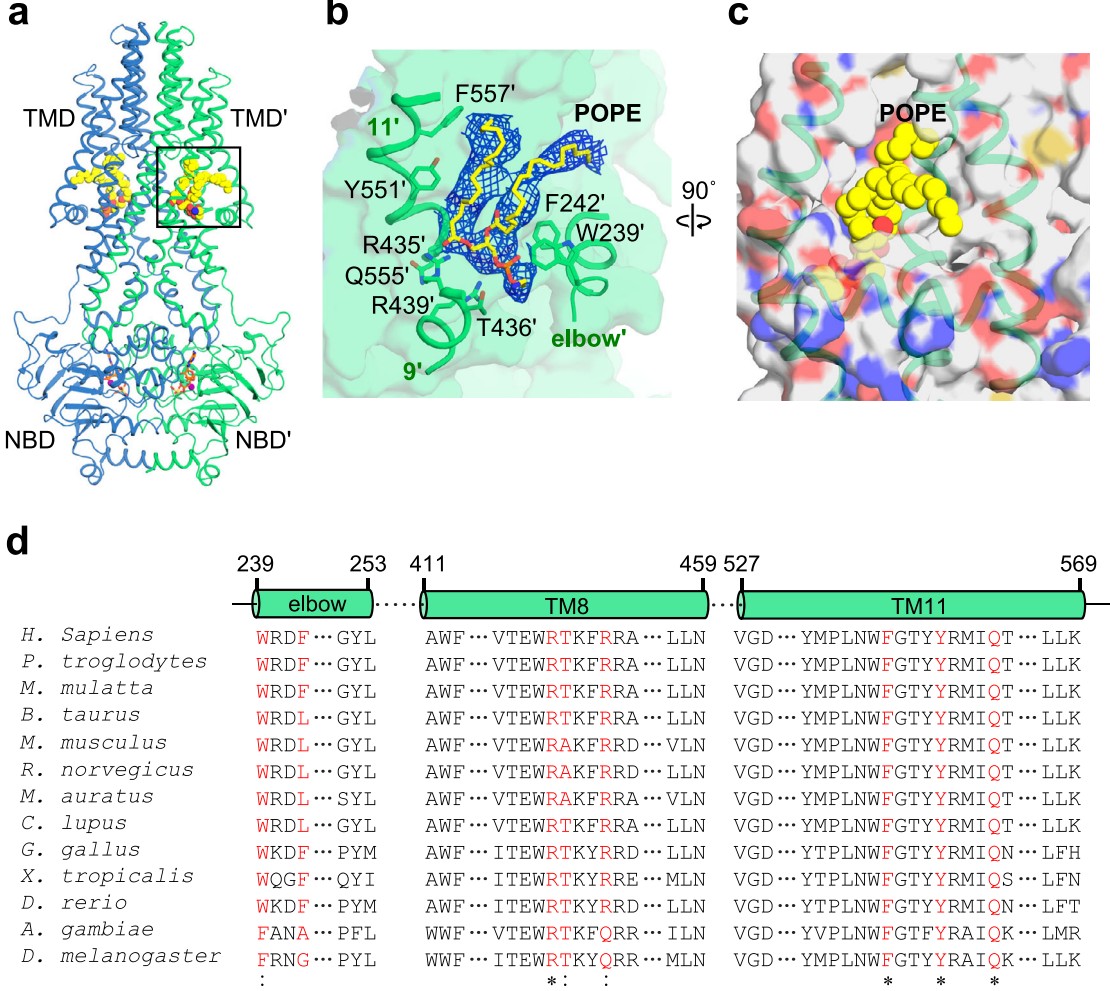

**Fig. 5 The post-occluded conformation of the hABCB6core-W546A mutant is stabilized by a phospholipid embedded within the hydrophobic groove of the TMD surface. a** Overall structure of the post-occluded W546A mutant in nanodiscs. The two monomers are colored blue and green, respectively. ADP·VO$_4$ is shown as sticks and Mg$^{2+}$ is shown as a magenta sphere. Bound POPE molecules are shown as yellow spheres. The area enlarged in (**b**) and (**c**) is boxed. **b** Close-up view of protein-POPE interactions. The POPE molecule is shown as yellow sticks and its EM density map (4 σ) is displayed as blue mesh. **c** Surface representation of the lipid-binding groove with the color scheme based on atom type. **d** Sequence alignment of ABCB6 homologs highlighting residue conservation (red) within the lipid-binding groove.

to the pre-occluded state rather than progressing directly to the outward-facing state[26]. This preference can be attributed, at least in part, to the high thermodynamic threshold between the inward-facing and outward-facing states under substrate-free conditions, as evidenced by the low ATPase activity of hABCB6core (Fig. 1a)[39]. Consequently, ATP-binding alone is likely insufficient to fully drive the complete formation of the outward-facing conformation.

## Discussion

Our findings, combined with previous cryo-EM data, show that ABCB6 adopts five major conformations during its transport cycle, namely apo inward-facing, substrate-bound inward-facing, nucleotide-bound pre- and post-occluded, and outward-facing conformations[24–26]. These results provide a more comprehensive characterization of the conformational cycle of ABCB6, which is governed by the intricate interplay between substrate binding/release, the ATPase cycle, and protein-lipid interactions (Fig. 5–7)[37]. Despite this advance in our understanding of ABCB6 dynamics, it remains challenging to explain why lipids affect differently the activities of hABCB6core and W546A mutant (Fig. 3). One possible explanation is that, in addition to their role

in stabilizing the transporter in specific conformational states (Fig. 5), lipids might also function as putative substrates. To expand our understanding of the interplay between ABCB6 and lipids, future research should employ a comprehensive approach that combines biochemical and biophysical experiments, cryo-EM studies, and computational simulations. Such an integrated approach would provide a deeper understanding of how lipids affect the functions and structure of ABCB6, including whether lipids indeed act as ABCB6 substrates. It would also throw light on why the W546A mutation renders the transporter unresponsive to lipid molecules (Fig. 3).

To the best of our knowledge, this study provides the first insights into the structural role of the stacking interactions of W546 aromatic rings and the binding of phospholipids within the TMD groove in governing the conformational preference between the outward-facing and post-occluded states of ABCB6. In detergent solution, the hABCB6core mostly adopted a post-occluded conformation in the presence of nucleotide and substrate (Fig. 8a). By contrast, the mutant transporter with only a single mutation at W546 (W546A) adopted mostly an outward-facing conformation under the same conditions (Fig. 8b). This observation suggests that, in detergent, the W546A mutation is sufficient to alter the energy landscape among conformations,

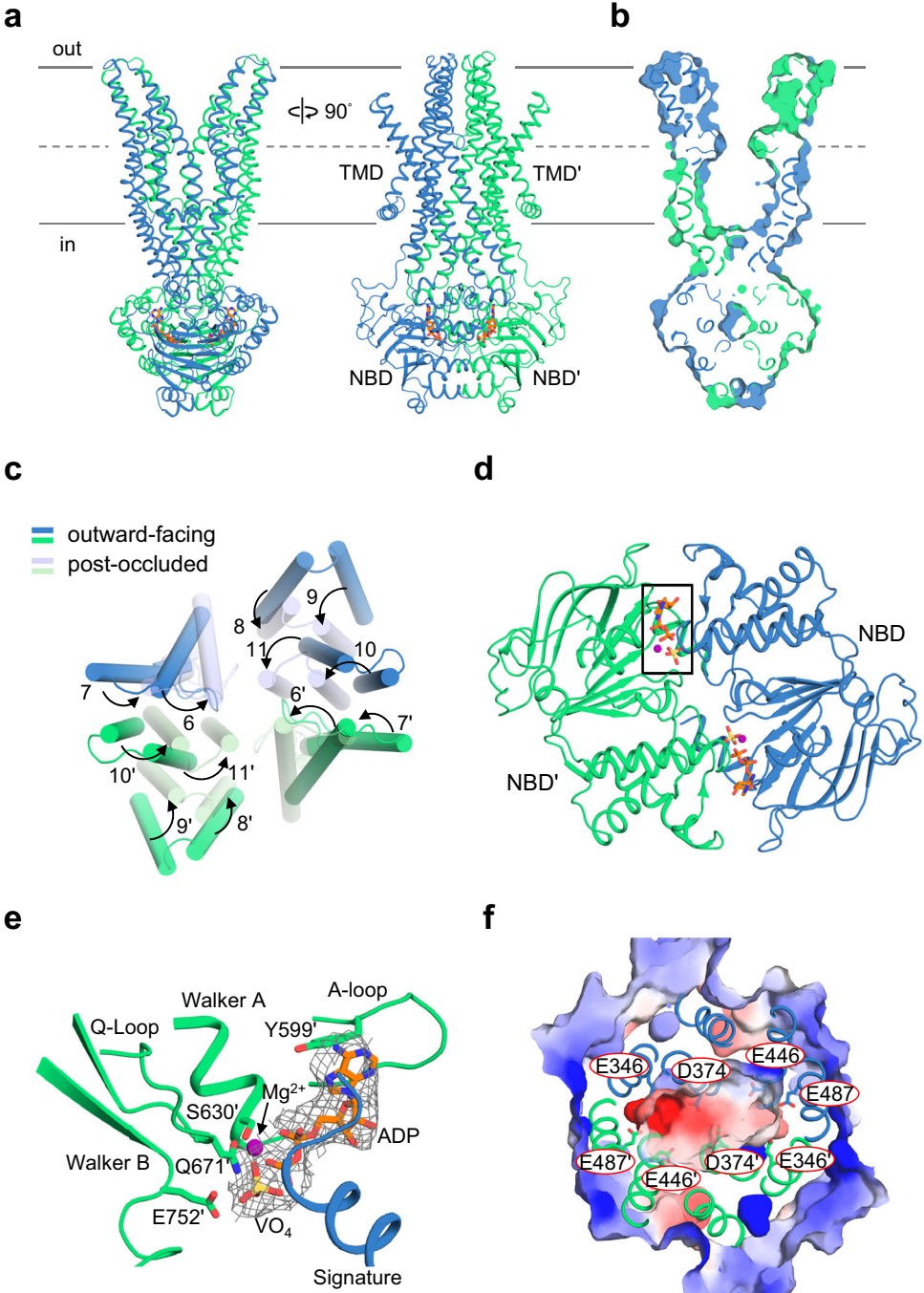

**Fig. 6 Cryo-EM structure of the hABCB6$^{core}$-W546A mutant in an outward-facing state. a** Overall structure of the W546A mutant in the outward-facing state. The two monomers are shown in blue and green, respectively. ADP·VO$_4$ is shown as sticks (orange and yellow) and Mg$^{2+}$ is shown as a sphere (magenta). **b** Surface slab view. **c** Comparison of the conformation of transmembrane helices between outward-facing and post-occluded states. **d** Bottom view of the NBD dimer. Mg$^{2+}$ and ADP·VO$_4$ are represented by spheres (magenta) and sticks (orange and yellow), respectively. The area enlarged in (**e**) is boxed. **e** Zoomed-in view of the ADP·VO$_4$-binding site. Side chains involved in ADP·VO$_4$ binding are shown as sticks. Cryo-EM map (gray mesh) of ADP·VO$_4$ is contoured at the 4 σ level. **f** Close-up view from the extracellular side into the transport pathway. The surface electrostatic potential, contoured from -5 kT/e (negative, red) to +5 kT/e (positive, blue) was represented using the APBS-PDB2PQR software suite, assuming a pH of 7.0 (http://www. poissonboltzmann.org). Overlaid structures of acidic residues are shown as sticks.

making the outward-facing state the most stable and lowest energy conformation. In the lipid environment, however, the transporter predominantly adopted the post-occluded state irrespective of the presence of the W546A mutation (Fig. 8c). Structural analysis suggests that the lipid inserted within the TMD groove may compensate for the W546A mutation by acting as a molecular wedge to resist conformational transitions to the

outward-facing form (Fig. 5). Thus, removal of the wedge-like lipid by harsh solubilization conditions would facilitate the transition to an outward-facing state.

These structural features may apply to other ABC transporters. For example, *Chaetomium thermophilum* Atm1 and *Arabidopsis thaliana* Atm3, orthologs of the ABCB7/HMT1/ABCB6 family, have aromatic residues (phenylalanine) at positions 396 and 435,

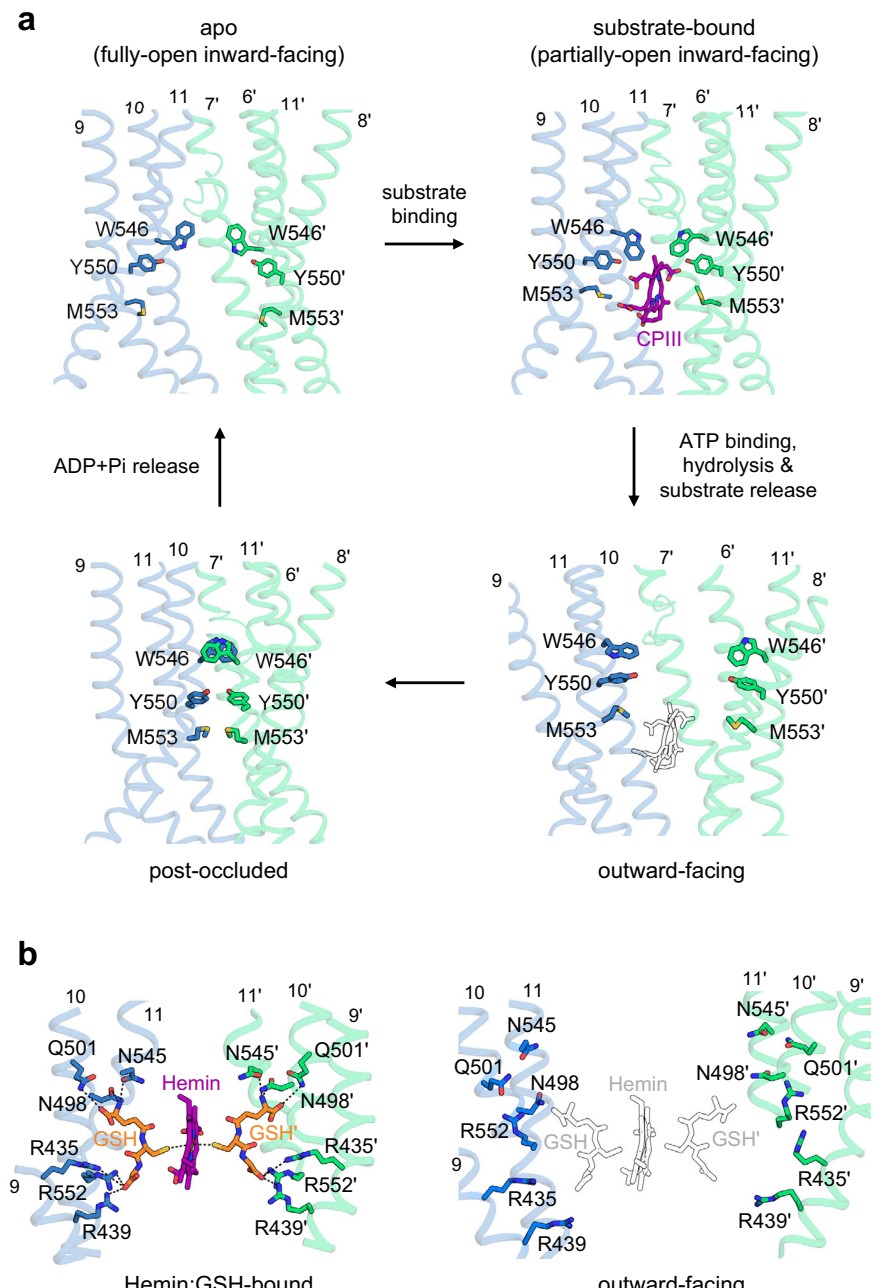

**Fig. 7 Conformational changes at the substrate-binding site during the transport cycle. a, b** Close-up views of the CPIII- (**a**) and hemin:GSH- **b** binding sites of each conformation viewed from within the plane of the membrane. Bound substrates and key residues that interact with substrates are represented by sticks and colored by heteroatom. The structures of CPIII and hemin:GSH superimposed on the outward-facing hABCB6core are drawn as empty black sticks.

respectively (Fig. 9a, b)[40,41]. These residues exert their effects to bring about complete occlusion of the transporter by stacking aromatic rings in a parallel-displaced geometry. Likewise, the zebrafish cystic fibrosis transmembrane conductance regulator (CFTR, also known as ABCC7) is stabilized in the occluded state via cation-π interactions between R353 and W1153 (Fig. 9c)[42]. Unlike other outward-facing ABC transporters, human P-glycoprotein (ABCB1) has only a small opening at the extracellular side of the membrane (Fig. 9d)[43]. In its structure, the two phenylalanines, F343 and F983, form a key structural contact via edge-to-face interactions that appear to prevent the wide opening of the extracellular segments. The *Escherichia coli* lipid transporter MsbA lacks aromatic residues in its translocation pathway[44], but its stability and activity depend on directly bound lipids[45–47]. Intriguingly, the cryo-EM structure of the ADP·VO4-bound occluded state of Salipro-reconstituted MsbA revealed a phospholipid-like density in a similar position to that observed in hABCB6core[48]. The generalizability of these findings to a broader range of ABC transporters cannot be guaranteed. However, our study provides valuable insights that can be utilized to devise strategies for trapping ABC transporters in desired conformations, thereby enabling more detailed structural and functional studies.

## Materials and methods

**Cloning, expression, and purification of hABCB6core and its variants.** The core region of human ABCB6 (GenBank accession number AB039371.2, residues 206−842) was cloned into the

## a

in detergent micelle (Figs. 2a, b)

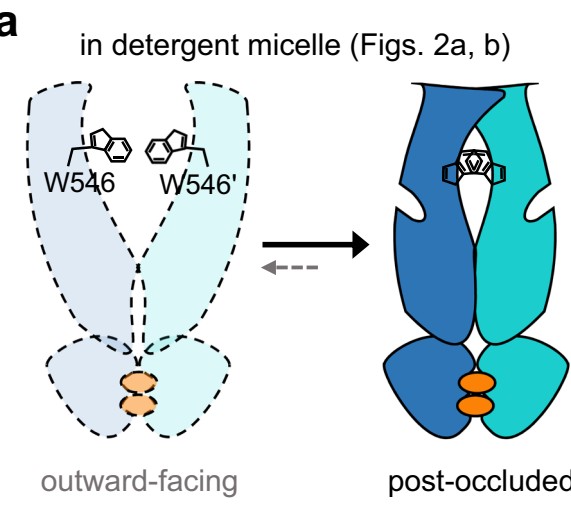

outward-facing post-occluded

## b

in detergent micelle (Figs. 4a, b)

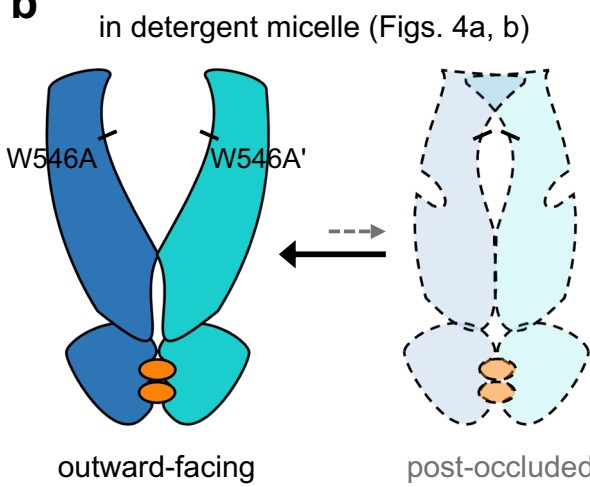

outward-facing post-occluded

## c

in lipid environment (Figs. 4c, d)

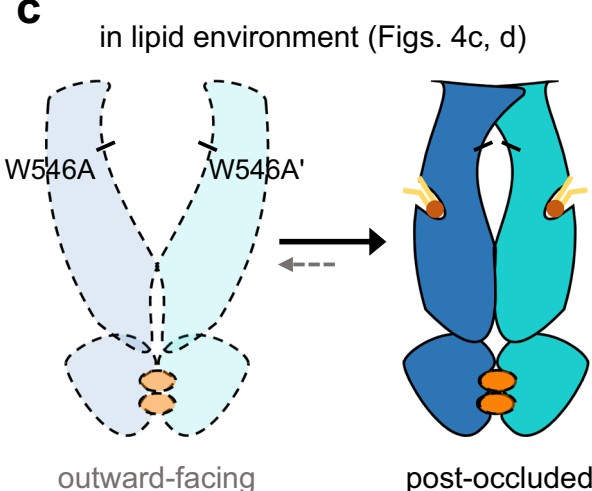

outward-facing post-occluded

**Fig. 8 Schematic diagrams of conformational equilibria between the outward-facing and post-occluded configurations of hABCB6^core and the W546A mutant. a–c** Subunits are colored blue and cyan. Mg$^{2+}$/ATP is depicted as an orange oval. Phospholipids are presented as brown heads and yellow tails. Dark-colored states indicate 'preferred' forms, while pale-colored states represent 'unfavorable' forms.

pVL1393 vector (BD Biosciences) with a C-terminal thrombin-cleavable enhanced green fluorescent protein (eGFP) and dec-ahistidine (10×His) tag[25]. Site-directed mutation (W546A and E752Q) was achieved by overlap PCR and confirmed by DNA sequencing. Baculovirus-infected High Five (Hi5) cells were cultured for 72 h at 28 °C. Harvested cells were resuspended in buffer containing 20 mM HEPES-NaOH pH 7.0, 200 mM NaCl, 1 mM phenylmethylsulfonyl fluoride (PMSF), 10 mM MgCl$_2$, and 40 μg/mL DNase I. Cells were broken by sonication at 40% amplitude with 5 s pulses separated by 6 s pauses for 3 min using a Branson Sonifier equipped with a 3.2 mm tip. Cell membranes were collected by ultracentrifugation at 300,000 g for 1 h. Proteins were solubilized for 2 h with 2% (w/v) n-dodecyl-β-D-maltopyranoside (DDM; Anatrace) and 0.2% (w/v) cholesteryl hemisuccinate (CHS; Anatrace). Non-solubilized material was removed by centrifugation at 300,000 g for 30 min and the supernatant was loaded onto an anti-GFP DARPin-based affinity column[49]. After washing the resin with buffer containing 20 mM HEPES-NaOH pH 7.0, 200 mM NaCl, 0.056% (w/v) 6-cyclohexyl-1-hexyl-β-D-maltoside (cymal-6; Anatrace), and 0.0056% (w/v) CHS, bound protein was eluted from the resin by on-column thrombin cleavage (Lee Biosolutions). Eluted protein was further purified by gel filtration chromatography using a Superdex 200 Increase 10/300 GL column (Cytiva). All purification steps were performed on ice or at 4 °C.

**Cloning, expression, and purification of hApoA-I (Δ1−43).** The gene encoding human apolipoprotein A-I (hApoA-I) spanning residues D44 to L243 (Δ1−43) was cloned into the pET21a vector (Novagen) containing a thrombin-cleavable hexahistidine (6×His) tag at the C-terminus[50]. Recombinant protein was produced in *Escherichia coli* BL21 (DE3) cells cultured in LB medium at 37 °C. When the OD$_{600}$ reached 0.6, 1 mM isopropyl-β-D-thiogalactopyranoside (IPTG, GoldBio) was added and cells were incubated for a further 5 h at 37 °C. After harvesting cells by centrifugation at 7700 g for 10 min, the pellet was resuspended in lysis buffer containing 20 mM Tris-HCl pH 8.0, 200 mM NaCl, 1 mM PMSF, and 10 μL/mL of DNase I. After sonication on ice, cell debris was removed by centrifugation at 30,000 g for 1 h. The supernatant was loaded onto Ni-NTA agarose resin (GoldBio), washed thoroughly with buffer containing 20 mM Tris-HCl pH 8.0, 200 mM NaCl, and 30 mM imidazole, and elution of bound proteins was performed using a gradient of imidazole up to 500 mM. The 6×His tag was removed by thrombin cleavage overnight and the protein was further purified by HiTrap Q (Cytiva) anion exchange and Superdex 200 Increase 10/300 GL gel filtration chromatography steps. All purification steps were performed at 4 °C.

**Nanodisc reconstitution and purification.** Lipids (Avanti Polar Lipids) used for nanodisc assembly were porcine polar lipid extract, POPC (1-palmitoyl-2-oleoyl-glycero-3-phosphocholine), POPS (1-palmitoyl-2-oleoyl-sn-glycero-3-phospho-L-serine), and POPG (1-palmitoyl-2-oleoyl-sn-glycero-3-phospho-(1'-rac-glycerol)). Each lipid dissolved in chloroform was dried under a gentle stream of nitrogen. The resulting thin lipid film was dissolved in buffer comprising 20 mM HEPES-NaOH pH 7.0, 200 mM NaCl, 0.056% (w/v) cymal-6, and 0.0056% (w/v) CHS. After water bath sonication at 22 °C for 1 min, the lipid solution was mixed with hApoA-I (Δ1−43) and ABCB6^core protein (or W546A mutant) at a molar ratio of 150:5:1 and incubated at 4 °C for 2 h. For detergent removal, the mixture was incubated with 200 mg of SM2 Bio-Beads (Bio-Rad) for 4 h at 4 °C. The Bio-Beads were replaced twice with 200 mg fresh beads and incubated for 16 h and 1 h in between. The final supernatant was loaded

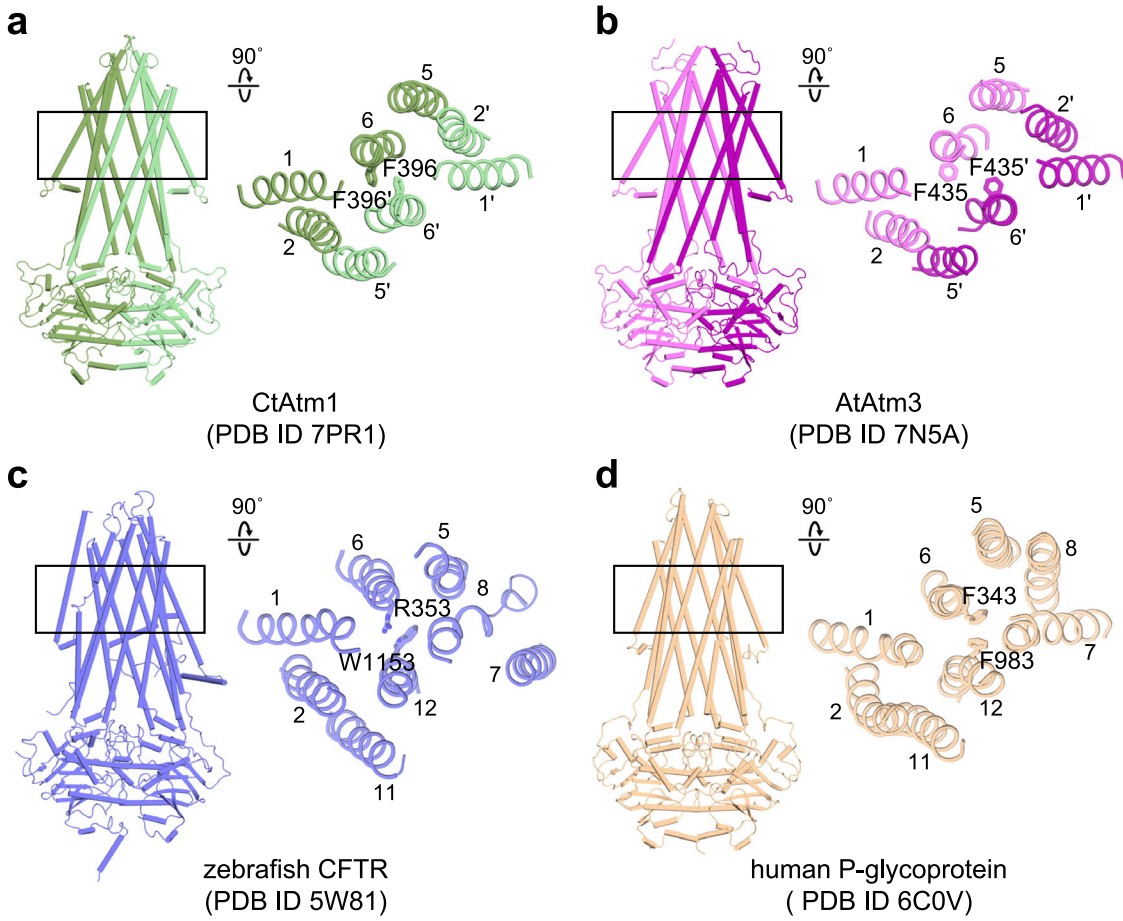

**Fig. 9 Role of conserved aromatic residues in stabilizing the occluded conformation of the ABC transporter family. a–d** Cylindrical (left) and cartoon (right) representations of representative ABC transporters. Residues involved in stabilization of the occluded conformation are shown as sticks. **a** *C. thermophilum* Atm1 (PDB ID 7PR1). **b** *A. thaliana* Atm3 (PDB ID 7N5A). **c** Zebrafish CFTR (ABCC7, PDB ID 5W81). **d** Human P-glycoprotein (ABCB1, PDB ID 6C0V).

onto a Superdex 200 Increase 10/300 GL column equilibrated with buffer comprising 20 mM HEPES-NaOH pH 7.0 and 200 mM NaCl to separate protein-loaded nanodiscs from empty nanodiscs.

**ATP hydrolysis assay.** ATP hydrolysis activity of purified proteins was measured by NADH-coupled ATPase assay as described previously[33]. Briefly, 10 μg of protein sample was added to 100 μL of ATPase reaction buffer containing 50 mM HEPES-KOH pH 8.0, 60 μg/mL pyruvate kinase, 32 μg/mL lactate dehydrogenase, 4 mM phosphoenolpyruvate, 0.3 mM NADH, 2 mM ATP, 10 mM $MgCl_2$, 0.056% (w/v) cymal-6, and 0.0056 % (w/v) CHS in the presence or absence of substrates. For proteins reconstituted into nanodiscs, cymal-6 and CHS were not added to the reaction buffer. ATPase activity was monitored at 37 °C every 10 s for 3 min by measuring the decrease in NADH absorbance at 340 nm ($\varepsilon M = 6220$ cm$^{-1}$ M$^{-1}$). Inhibition of ATP hydrolysis was measured by incubating the protein with 2 mM ATP plus 2 mM sodium orthovanadate, 8 mM NaF/2 mM $AlCl_3$, or 2 mM $Na_2SO_4$ for 2 h on ice. Basal ATPase activity was measured with an ATP gradient up to 2 mM. Kinetic parameters for ATP hydrolysis were determined using the Michaelis-Menten equation in GraphPad Prism 7.0 (GraphPad software).

**Proteoliposome reconstitution.** Liposomes were prepared using a 1:1 (w/w) ratio of Egg PC and *E. coli* polar lipid extract (Avanti Polar Lipids). Lipids solubilized in chloroform were dried under

nitrogen gas and resuspended in buffer containing 20 mM HEPES-NaOH pH 7.0 and 200 mM NaCl. After five freeze-thaw cycles, the liposome suspension was passed through a Mini-Extruder equipped with a 200 nm polycarbonate membrane filter (Avanti Polar Lipids) to produce unilamellar liposomes. Liposomes were destabilized with 5 mM n-Decyl-β-D-Maltopyranoside (DM; Anatrace) at a 1:1 (mol/mol) ratio for 3 h at 22 °C. The purified protein was added to the mixture at a 1:50 (w/w) protein to lipid ratio and incubated for 2 h at 4 °C with gentle shaking. Detergent was removed by adding Bio-Beads (200 mg/mL) for 2 h at 4 °C followed by overnight incubation with fresh Bio-Beads. The supernatant was further treated with fresh Bio-Beads for 2 h, and proteoliposomes were separated by Histodenze density gradient centrifugation (35%: 25%: 0%; Sigma-Aldrich) at 259,000 g for 1 h at 4 °C[51]. The resulting proteoliposomes were diluted in buffer comprising 20 mM HEPES-NaOH pH 7.0 and 200 mM NaCl to a final phospholipid concentration of 0.5 mM. Liposomal lipid concentrations were determined by colorimetric phosphate assay[52].

**CPIII transport assay.** The CPIII transport activities of hABCB6$^{core}$ and its variants were measured as described in a previous study[25]. Briefly, 60 μL of proteoliposomes (~1.8 μg protein) were added to 240 μL reaction buffer containing 50 mM HEPES-KOH pH 8.0, 70 mM KCl, 50 μM CPIII, 10 mM $MgCl_2$, and 4 mM ATP (or its analogs). After incubation at 37 °C for 30 min, proteoliposomes were pelleted by ultracentrifugation at

**Table 1 Cryo-EM data collection, refinement and validation statistics.**

| | W546A in nanodisc post-occluded (EMDB-36937) (PDB 8K7B) | W546A in detergent outward-facing (EMDB-36938) (PDB 8K7C) | hABCB6$^{core}$ in detergent post-occluded |
|---|---|---|---|
| **Data collection and processing** | | | |
| Magnification | 100,000 | 105,000 | 100,000 |
| Voltage (kV) | 200 | 300 | 200 |
| Electron exposure (e⁻/Å²) | 40 | 50 | 40 |
| Defocus range (μm) | -0.7 ~ -2.4 | -0.7 ~ -2.4 | -0.7 ~ -2.4 |
| Pixel size (Å) | 0.830 | 0.851 | 0.830 |
| Symmetry imposed | C2 | C1 | C2 |
| Initial particle images (no.) | 1,192,936 | 1,008,231 | 673,626 |
| Final particle images (no.) | 323,588 | 56,293 | 137,035 |
| Map resolution (Å) | 3.9 | 3.9 | 3.8 |
| FSC threshold | 0.143 | 0.143 | 0.143 |
| Map resolution range (Å) | 3.5 ~ 6.0 | 3.5 ~ 8.0 | - |
| **Refinement** | | | |
| Initial model used (PDB code) | 7EKL | 7EKL | |
| Model resolution (Å) | 3.5 | 3.5 | |
| FSC threshold | 0.143 | 0.143 | |
| Model resolution range (Å) | 2.4 ~ 4.0 | 2.4 ~ 4.0 | |
| Map sharpening $B$ factor (Å²) | 237.8 | 153.4 | |
| Model composition | | | |
| Non-hydrogen atoms | 9,332 | 9,205 | |
| Protein residues | 164 | 54 | |
| Ligands | | | |
| $B$ factors (Å²) | | | |
| Protein | 69.4 | 119.7 | |
| Ligand | 23.9 | 103.1 | |
| R.m.s. deviations | | | |
| Bond lengths (Å) | 0.008 | 0.009 | |
| Bond angles (°) | 1.196 | 0.925 | |
| Validation | | | |
| MolProbity score | 2.3 | 2.1 | |
| Clashscore | 17.5 | 11.5 | |
| Poor rotamers (%) | 0.8 | 0.1 | |
| Ramachandran plot | | | |
| Favored (%) | 88.9 | 91.5 | |
| Allowed (%) | 11.1 | 8.5 | |
| Disallowed (%) | 0 | 0 | |

210,000 g for 1 h at 4 °C, then washed three times with buffer containing 50 mM HEPES-KOH pH 8.0, 70 mM KCl, and 10 mM MgCl$_2$. The resulting proteoliposomes were lysed by adding 10% (w/v) sodium dodecyl sulfate (SDS; final concentration 2%). Transport of CPIII was measured as the fluorescence intensity using a microplate reader at an excitation wavelength of 395 nm and an emission wavelength of 610 nm. Empty liposomes were used as a negative control.

**Cryo-EM grid preparation and data collection**. Before grid preparation, proteins (6 mg/mL for detergent samples or 0.4 mg/mL for nanodisc samples) were incubated with 100 μM CPIII, 2 mM MgCl$_2$, 2 mM ATP, and 2 mM sodium orthovanadate (Na$_3$VO$_4$) for 30 min at 22 °C. Next, 3 μL aliquots of samples were applied to freshly glow-discharged gold grids covered with holey carbon film (300-mesh Au R1.2/1.3; Quantifoil). The grids were blotted for 2–3 s (blot force 2–3) at 4 °C with 100% humidity and plunge-frozen in liquid ethane using a Vitrobot Mark IV (Thermo Fisher Scientific). Detergent-purified hABCB6$^{core}$ and nanodisc-reconstituted W546A grids were imaged using a 200 kV Talos Arctica transmission electron microscope (Thermo Fisher Scientific) equipped with a K3 detector and a GIF BioQuantum energy filter (slit width 20 eV; Gatan). The W546A sample in detergent micelles was loaded

onto a 300 kV Titan Krios transmission electron microscope (Thermo Fisher Scientific) equipped with a K3 detector. Each dataset was collected using EPU software in standard counting mode. The EM data collection parameters for each sample are summarized in Table 1.

**Cryo-EM data processing**. Three datasets comprising 10,280, 8477, and 6833 micrographs were acquired for hABCB6$^{core}$ and W546A in detergent micelles, and W546A in nanodiscs, respectively. Similar processing strategies were applied to detergent-purified samples of hABCB6$^{core}$ and W546A. Briefly, movie stacks were corrected for beam-induced motion and dose-weighted using MotionCor2[53]. The resulting micrographs were subjected to patch-based contrast transfer function (CTF) estimation by Gctf v1.18[54]. After discarding low-quality micrographs, a subset of images was randomly selected and particles were autopicked using the Laplacian-of-Gaussian filter in RELION 4.0[55]. Picked particles were extracted using a box size of 230×230 pixels and subjected to 2D classification. Good 2D class averages were used as templates for reference-based autopicking of all micrographs. After multiple rounds of 2D classification, particles comprising the best 2D classes were selected to generate an initial model for 3D reconstruction, followed by 3D classification. Particles displaying typical features of ABC transporters

were pooled and subjected to 3D refinement[56]. For the detergent-purified W546A dataset, local CTF refinement and Bayesian polishing were additionally performed to improve CTF estimation by adjusting the per-particle defocus values and high-order CTF terms[56]. The resulting particles were then exported to CryoSPARC v3.3.2[35] for homogeneous 3D refinement, followed by non-uniform refinement[57]. The final resolution of the W546A protein (3.9 Å using C1 or C2 symmetry) was estimated using a gold-standard Fourier shell correlation (FSC) cut-off criterion of 0.143[58]. A local resolution map was calculated from the two half-maps using CryoSPARC.

The dataset for the nanodisc-reconstituted W546A mutant was processed using CryoSPARC. Raw images underwent patch-based motion correction and CTF estimation. Micrographs with low quality, characterized by low resolution, high defocus, low figure of merit, and high astigmatism, were excluded from further processing. The complete set of micrographs was used for automated particle picking using a blob picker, and particles were extracted with a box size of 256 pixels and subjected to 2D classification. Multiple rounds of 2D classification were performed, and particles corresponding to the best 2D classes were chosen to generate an ab initio model for 3D reconstruction. Subsequently, 3D heterogeneous and homogeneous refinement steps were employed, followed by application of global CTF refinement to further enhance map quality. The final 3D reconstruction was accomplished through non-uniform refinement with either C1 or C2 symmetry, yielding a resolution of 3.9 Å.

**Model building and refinement**. To generate a starting model, we docked an atomic model of pre-occluded hABCB6[core] (PDB ID 7EKL)[26] into the EM density map of W546A using UCSF Chimera[59]. The position and orientation of the coordinate were initially adjusted using rigid body refinement and morphing in the phenix.real_space_refine program of the PHENIX suite[60]. The model was further edited and refined by iterative rounds of manual model building in Coot[61] and real-space refinement in PHENIX. Amino acids in poor density regions were built as poly-alanine. The rotameric state of the W546 side chain was computed using the SWISS-MODEL server (http://swissmodel.expasy.org). The quality of the final structure was validated by MolProbity[62]. Refinement and validation statistics for the W546A protein are summarized in Table 1. All molecular graphics were prepared using PyMOL (https://pymol.org/2/), UCSF Chimera[59], and Chimera X[63].

**Reporting summary**. Further information on research design is available in the Nature Portfolio Reporting Summary linked to this article.

## Data availability
Refined atomic coordinates for the hABCB6[core]-W546A mutant in the post-occluded and outward-facing states have been deposited in the Protein Data Bank (PDB) under accession codes 8K7B and 8K7C, respectively. Cryo-EM density maps have been deposited in the Electron Microscopy Data Bank (EMDB) under accession codes EMD-36937 and EMD-36938, respectively. Source data for the graphs and charts in the main figures is given as Supplementary Data 1 and any remaining information can be obtained from the corresponding author upon reasonable request.

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

## Acknowledgements

We thank Dr. Jae-Woo Ahn and Mrs. Su Jeong Kim of POSTECH for helping with EM data collection. This research was supported by grants from the National Research Foundation (NRF) funded by the Ministry of Science, ICT, and Future Planning of Korea (NRF-2019M3E5D6063908, NRF-2021M3A9I4022846, and NRF-2022R1A2C1091278), and by a grant from the GIST Research Institute (GRI) IIBR funded by GIST in 2023.

## Author contributions

S.S.L. and M.S.J. designed the experiments. S.S.L. performed protein purification with support from S.H.C.. S.S.L. conducted biochemical experiments with the help of J.G.P. and E.J.. S.S.L. carried out cryo-EM experiments with the assistance of S.K.. J.W.K. granted access to cryo-EM equipment. S.S.L. prepared figures and M.S.J. wrote the manuscript with the help of all authors.

## Competing interests

The authors declare no competing interests.
