## [Peer Review File · Communications Biology]

Reviewers' comments:

Reviewer #1 (Remarks to the Author):

In this paper the authors follow up on their earlier work to present the cryo-EM structure of ABCB6 in an outward conformation. Recent years have witnessed a spate of ABC structures, which have allowed classification of the various ABC-folds, highlighting commonalities between different ABC transporters. This structure is very similar to known outward-facing Type IV structures. However, new structural information, if interpreted correctly, may shed light on the function of ABCB6.

1. Recent findings have questioned the mitochondrial localization of ABCB6, and the initial notion of ABCB6-mediated mitochondrial porphyrin transport is also heavily debated. Recent reviews on ABCB6 detailing this controversy should be cited (<https://febs.onlinelibrary.wiley.com/doi/full/10.1002/1873-3468.13967>).
2. Was the full-length protein also studied? Even if results were negative, please report them.
3. The authors argue that the population of inward-facing molecules in the presence of vanadate indicates that "opening of the extracellular part of the TMD may be a short-lived and high-energy state of the hABCB6core transport cycle." However, this explanation contradicts the finding that in the presence of vanadate, the ABCB6 ATPase is fully inhibited, which indicates that all ABCB6 molecules are stably trapped by vanadate. To this reviewer a more plausible interpretation would be to conclude that the inward-facing molecules that persist in the presence of vanadate are transport-incompetent. That the experimental conditions influence transport activity/competence, is also shown by the different results obtained in detergent and nanodiscs.
4. This is relevant also from the perspective of the main conclusion of the study, which postulates that "the highly conserved residue W546 influences conformational equilibria."
5. Based on the authors' earlier paper, deletion of the TM 7 loop supposedly destabilizes the inward-facing conformation, thereby facilitating the transition to the outward-facing conformation. Was this verified?
6. The authors seem to suggest that ABCB6 has several interdomain interactions that prevent the adoption of the outward-facing state. Do they stand by this interpretation? Is ABCB6 unique in that regard- especially considering that other ABC transporters can be readily trapped in an outward facing conformation in the presence of vanadate?

Minor comments:

Typo: coloboma (not colobama)

Page 5, line 92: block (not blocks)

Reviewer #2 (Remarks to the Author):

In this paper, Lee and colleagues study a homodimeric ABC pump, ABCB6, that translocates porphyrin from the cytoplasm into mitochondria for heme biosynthesis. It was previously shown by Song and colleagues (Cell Discovery (2021) 7:55) that a conserved Trp (W546 in each monomer) stabilizes an ATP-bound occluded conformation of an ATPase inactive E-to-Q mutant. Here, Lee and colleagues replaced this Trp residue by an Ala (W546A) and they solved its 3D structure by Cryo-EM. This mutant, in the presence of ATP/Vi, is capable to trap ADP/Vi in the ATP-binding sites at the interface of the two NBDs, thereby adopting preferentially an outward-facing conformation prone to porphyrins release.

Although the results obtained with this mutant are quite informative as they reveal a new outward-facing structure for ABCB6, possibly allowing porphyrin release, the catalytic cycle proposed in Figure

5 does not make sense. Indeed, the previous structure obtained by Song et al. and used in this model of the catalytic cycle corresponds to a prehydrolytic state (ATP-bound), as also stated here by the authors (line 85: "This knowledge allowed us to hypothesize that such a pre-hydrolytic state effectively traps hABC6core in the occluded configuration..."). In contrast, the outward-facing structure reported here has been obtained by vanadate trapping. This requires ATP hydrolysis to take place because, as shown before by Senior's team, you need first to hydrolyse ATP, then Pi leaves the NBD and Vi takes its place before ADP leaves the site. So the reaction is $ATP \rightarrow ADP + Pi$ \rightarrow Pi leave \rightarrow Vi comes into the Pi site \rightarrow ADP/Vi remains tightly bound to the ATP site and inhibits the ATPase activity. Therefore, the OF structure obtained here reflects a post-hydrolytic state (or possibly a transition state) but does not precede the ATPase activity. Thus, this structure cannot occur before the ATP bound occluded conformation in the catalytic cycle as proposed in the model in Figure 5.

Possibly, the ATP-bound occluded state observed before is a way to prevent basal ATPase activity in the absence of porphyrin (with the two Trp closing the gate) and this might explain the increase in ATP hydrolysis when the mutation W564A is created. Anyway, it has to occur before the ADP/Vi bound state in the catalytic cycle. Therefore, the authors should consider another model for the catalytic cycle as the one they proposed here is quite confusing and does not fit with a lot of Biochemical data obtained on different ABC transporters.

Reviewer #3 (Remarks to the Author):

1. The ATPase assays reported in Figure 1 (and elsewhere), were conducted in detergent solution. The authors demonstrate later that lipids have an important role in determining the conformational equilibria, questioning the relevance of the kinetic parameters determined in detergent. I highly recommend that the authors repeat these experiments in liposomes or nanodiscs.

2. "...capping the binding site from the extracellular space to sequester substrates in the central cavity (Figs. 2c, d) "

It is unclear where this "central cavity" is, and where is substrate sequestered. This should be made absolutely clear and visualised with HOLE if the substrate bound structure is unknown

3. "Furthermore, this highly conserved face-to-face aromatic interaction (Fig. 2e) appears to contribute to reaching a stable occluded conformation"

How is this concluded, and why would the disruption of this stacking specifically favor the outward facing state since in the inward facing state this interaction also does not exist.

4. The kinetic values reported here for the W546A mutant substantially differ from those previously reported by the authors. This should be mentioned and explained.

5. "We propose two potential explanations: (i) membrane lipids may alter the conformational preferences of the transporter by interacting with its specific structural motifs; (ii) our nanodisc system may not have sufficient elasticity or flexibility to allow the TMDs to open to the extracellular side"

This statement gives the impression as if the detergent solution is the better mimic and that the nanodiscs introduce artefacts. This is a very strange conclusion, since it is broadly accepted that detergents allow a greater conformational flexibility than lipid mimics, as was observed with many ABC transporters. The native conformational dynamics of ABCB6 seems to be dominated by the IF conformation with very brief excursions to the OF one. Combining the mutation and the detergent shifts the native conformational equilibria such that the OF conformation is either more frequently visited or is longer lived. This issue requires clarification.

6. "To confirm the proposed role of W546 as a structural stabilizer in the occluded state, we generated the W546A mutant and compared its functional properties with hABCB6" This mutant was previously generated and characterised by the authors. Please clarify

7. " Comparison of the outward-facing and occluded states revealed that the NBD structure and nucleotide-binding mode are nearly identical despite differences in the topological arrangement of the TMDs"

Fig 4C shows the exact opposite: there are considerable differences in the conformation of the NBDs between the two states. Although it is not shown, I imagine that there are also considerable differences in the positions of the catalytic residues. These differences need to be mentioned and discussed in the context of the proposed transport mechanism

8. "However, a single mutation of W546 to alanine is sufficient to alter the energy landscape among conformations, making the outward-facing state the most stable and lowest energy (Fig. 6b)"

While this is true in detergent solution, this is not the case in nanodisks, broadly considered to be a better a membrane-mimetic environment. How is this fact reflected in the proposed mechanism?

Reviewer #4 (Remarks to the Author):

In this paper, the authors determined the outward-facing transient state of human ATP-binding cassette transporter 6 of W546A mutant using cryo-EM. Previously, the occluded structures and inward structures were determined using cryo-EM and X-ray structures. This is the first study to determined the outward-facing structure thanks to the W546A mutant, where W546 stabilizes the occluded structures significantly.

The structural results have high impacts and the paper including figures are well written. The proposed mechanisms on the role of W546 are likely and have general importance comparing to other ABC transporters as shown in Figure 6 of the main text.

What I am concerned is the large effect on detergent micelles and nanodisks in Figure 3. Though the same mutant is used, outward-facing structures are dominant in micelles, while inward-facing structures are dominant in nanodisk. My question is how the latter structures are stabilize without W546. Is it possible to show atomistic structures of the inward-facing form in nanodisk?

Related question is that if you change the lipid components in nanodisk, is the structure changed from inward-facing/occluded states to outward-facing structures? I would like to hear more clear explanations about how lipid molecules can replace the interaction from W546, if possible.

Computational works, for instance, molecular modeling and simulation, might be useful for this purpose. I don't request the authors to include such computational works in the paper, since it would take more time and effort before the publication. However, it would be useful to discuss such possibility in Discussion section and to attempt such simulations in the later works.

Comments from Reviewers (Bold) and Author Responses

Reviewer #1 (Remarks to the Author):

In this paper the authors follow up on their earlier work to present the cryo-EM structure of ABCB6 in an outward conformation. Recent years have witnessed a spate of ABC structures, which have allowed classification of the various ABC-folds, highlighting commonalities between different ABC transporters. This structure is very similar to known outward-facing Type IV structures. However, new structural information, if interpreted correctly, may shed light on the function of ABCB6.

1. Recent findings have questioned the mitochondrial localization of ABCB6, and the initial notion of ABCB6-mediated mitochondrial porphyrin transport is also heavily debated. Recent reviews on ABCB6 detailing this controversy should be cited (<https://febs.onlinelibrary.wiley.com/doi/full/10.1002/1873-3468.13967>).

>> We have revised the text and cited the reference as suggested (lines 46–49).

2. Was the full-length protein also studied? Even if results were negative, please report them.

>> In the revised text, we have mentioned the results of the previous structural analysis of full-length human ABCB6. Please see lines 63–68.

3. The authors argue that the population of inward-facing molecules in the presence of vanadate indicates that “opening of the extracellular part of the TMD may be a short-lived and high-energy state of the hABCB6core transport cycle.” However, this explanation contradicts the finding that in the presence of vanadate, the ABCB6 ATPase is fully inhibited, which indicates that all ABCB6 molecules are stably trapped by vanadate. To this reviewer a more plausible interpretation would be to conclude that the inward-facing molecules that persist in the presence of vanadate are transport-incompetent. That the experimental conditions influence transport activity/competence, is also shown by the different results obtained in detergent and nanodiscs.

>> We agree with this comment and have revised the text, as suggested (lines 113–119).

This is relevant also from the perspective of the main conclusion of the study, which postulates that “the highly conserved residue W546 influences conformational equilibria.”

>> This section has been re-written to include additional experimental results demonstrating the importance of phospholipids, along with residue W546, in regulating the conformational preferences of ABCB6. Please see lines 169–197.

4. Based on the authors' earlier paper, deletion of the TM 7 loop supposedly destabilizes the inward-facing conformation, thereby facilitating the transition to the outward-facing conformation. Was this verified?

>> In our previous study (*Kim et al., Molecules and Cells 2022, 45(8), 513-602*), we investigated the impact of various deletion mutants targeting the TM7 bulge loop of hABCB6^{orc} on NADH-coupled ATPase activity. Notably, we found that complete deletion of the TM7 loop resulted in a significant (6.6-fold) increase in basal activity compared with the wild-type protein. This suggests that deletion of the TM7 loop may promote the formation of the outward-facing conformation by destabilizing the inward-facing conformation. Further examination of the structural and biochemical characteristics of the TM7 deletion mutant is necessary to unravel the intricate relationship between the TM7 loop and the conformational dynamics of ABCB6.

5. The authors seem to suggest that ABCB6 has several interdomain interactions that prevent the adoption of the outward-facing state. Do they stand by this interpretation? Is ABCB6 unique in that regard- especially considering that other ABC transporters can be readily trapped in an outward facing conformation in the presence of vanadate?

>> To the best of our knowledge, this is the first study highlighting the critical role of the W546 residue and direct protein-lipid interactions in stabilizing the occluded conformation of ABCB6. Consistent with this observation, we found that simultaneous disruption of these two stabilizing factors promotes the adoption of outward-facing forms. However, it is important to note that these factors may not be universally applicable to all ABC transporters. Unlike ABCB6, many ABC transporters (e.g., ABCB1, ABCB9, and ABCD1) can readily become trapped in an outward-facing conformation, NOT occluded, in the presence of ATP or its mimics (*Kim et al., Science 2018, 359(6378), 915-919; Park et al., Nat Comm 2022, 13, 5851; Le et al., Communications biology 2022, 5, 7*), indicating the existence of varying conformational dynamics and regulatory mechanisms among different ABC transporters. We have revised the text (lines 307–308) to clarify this point.

Minor comments:

Typo: coloboma (not colobama)

Page 5, line 92: block (not blocks)

>> Sorry for this mistake. It has been corrected. Thank you for pointing it out.

=====

Reviewer #2 (Remarks to the Author):

In this paper, Lee and colleagues study a homodimeric ABC pump, ABCB6, that translocates porphyrin from the cytoplasm into mitochondria for heme biosynthesis. It was previously shown by Song and colleagues (*Cell Discovery* (2021) 7:55) that a conserved Trp (W546 in each monomer) stabilizes an ATP-bound occluded conformation of an ATPase inactive E-to-Q mutant. Here, Lee and colleagues replaced this Trp residue by an Ala (W546A) and they solved its 3D structure by Cryo-EM. This mutant, in the presence of ATP/Vi, is capable to trap ADP/Vi in the ATP-binding sites at the interface of the two NBDs, thereby adopting preferentially an outward-facing conformation prone to porphyrins release.

Although the results obtained with this mutant are quite informative as they reveal a new outward-facing structure for ABCB6, possibly allowing porphyrin release, the catalytic cycle proposed in Figure 5 does not make sense. Indeed, the previous structure obtained by Song et al. and used in this model of the catalytic cycle corresponds to a prehydrolytic state (ATP-bound), as also stated here by the authors (line 85: “This knowledge allowed us to hypothesize that such a pre-hydrolytic state effectively traps hABCB6core in the occluded configuration...”). In contrast, the outward-facing structure reported here has been obtained by vanadate trapping. This requires ATP hydrolysis to take place because, as shown before by Senior’s team, you need first to hydrolyse ATP, then Pi leaves the NBD and Vi takes its place before ADP leaves the site. So the reaction is $ATP \rightarrow ADP + Pi$ \diamond Pi leave \diamond Vi comes into the Pi site \diamond ADP/Vi remains tightly bound to the ATP site and inhibits the ATPase activity. Therefore, the OF structure obtained here reflects a post-hydrolytic state (or possibly a transition state) but does not precede the ATPase activity. Thus, this structure cannot occur before the ATP bound occluded conformation in the catalytic cycle as proposed in the model in Figure 5.

>> We agree with this comment and have modified Figure 6 to incorporate the post-occluded conformation determined in this study.

Possibly, the ATP-bound occluded state observed before is a way to prevent basal ATPase activity in the absence of porphyrin (with the two Trp closing the gate) and this might explain the increase in ATP hydrolysis when the mutation W564A is created. Anyway, it has to occur before the ADP/Vi

bound state in the catalytic cycle. Therefore, the authors should consider another model for the catalytic cycle as the one they proposed here is quite confusing and does not fit with a lot of Biochemical data obtained on different ABC transporters.

>> We have modified Figure 6 as suggested.

=====

Reviewer #3 (Remarks to the Author):

1. The ATPase assays reported in Figure 1 (and elsewhere), were conducted in detergent solution. The authors demonstrate later that lipids have an important role in determining the conformational equilibria, questioning the relevance of the kinetic parameters determined in detergent. I highly recommend that the authors repeat these experiments in liposomes or nanodiscs.

>> As suggested, we measured basal ATPase activities using nanodisc-reconstituted proteins to examine how the lipid environment affects their kinetic parameters. These results are shown in Supplementary Figure 16 and discussed in lines 272–285.

2. “.....capping the binding site from the extracellular space to sequester substrates in the central cavity (Figs. 2c, d) “

It is unclear where this “central cavity” is, and where is substrate sequestered. This should be made absolutely clear and visualized with HOLE if the substrate bound structure is unknown.

>> As suggested, we have modified Figure 2c to indicate the position of the central cavity.

3. “Furthermore, this highly conserved face-to-face aromatic interaction (Fig. 2e) appears to contribute to reaching a stable occluded conformation in the presence of ATP.”

How is this concluded, and why would the disruption of this stacking specifically favor the outward facing state since in the inward facing state this interaction also does not exist.

>> Based on a detailed examination of the inward-facing and post-occluded structures of the W546A mutant, we have revised this section to include new findings highlighting the significance of direct interactions between protein and lipids for the stabilization of these two states, even in the absence of the W546 residue. Please see lines 175–197 and Figure 4.

4. The kinetic values reported here for the W546A mutant substantially differ from those

previously reported by the authors. This should be mentioned and explained.

>> As mentioned, the kinetic values obtained for the W546A mutant in our experimental conditions show some differences from previously reported values. We have discussed the potential reasons for these differences. Please see lines 142–145.

5. “We propose two potential explanations: (i) membrane lipids may alter the conformational preferences of the transporter by interacting with its specific structural motifs; (ii) our nanodisc system may not have sufficient elasticity or flexibility to allow the TMDs to open to the extracellular side”

This statement gives the impression as if the detergent solution is the better mimic and that the nanodiscs introduce artefacts. This is a very strange conclusion, since it is broadly accepted that detergents allow a greater conformational flexibility than lipid mimics, as was observed with many ABC transporters. The native conformational dynamics of ABCB6 seems to be dominated by the IF conformation with very brief excursions to the OF one. Combining the mutation and the detergent shifts the native conformational equilibria such that the OF conformation is either more frequently visited or is longer lived. This issue requires clarification.

>> We agree that this sentence is confusing and have re-written the text to clarify (lines 169–197).

6. “To confirm the proposed role of W546 as a structural stabilizer in the occluded state, we generated the W546A mutant and compared its functional properties with hABCB6” This mutant was previously generated and characterized by the authors. Please clarify.

>> We have revised this sentence to clarify the point (lines 137–139).

7. “ Comparison of the outward-facing and occluded states revealed that the NBD structure and nucleotide-binding mode are nearly identical despite differences in the topological arrangement of the TMDs”

Fig 4C shows the exact opposite: there are considerable differences in the conformation of the NBDs between the two states. Although it is not shown, I imagine that there are also considerable differences in the positions of the catalytic residues. These differences need to be mentioned and discussed in the context of the proposed transport mechanism.

>> It appears that there may have been a misunderstanding by the reviewer regarding the content of Figure 4C. We would like to clarify that Figure 4C actually presents a structural comparison of the α -helical TMDs, NOT the NBDs, between the outward-facing and post-occluded states. To provide further

clarification, we have included a close-up view of the nucleotide-binding site of NBDs showing the comparable positions of catalytic residues between the two states (Supplementary Figure 15).

8. “However, a single mutation of W546 to alanine is sufficient to alter the energy landscape among conformations, making the outward-facing state the most stable and lowest energy (Fig. 6b)”

While this is true in detergent solution, this is not the case in nanodiscs, broadly considered to be a better a membrane-mimetic environment. How is this fact reflected in the proposed mechanism?

>> We agree that this sentence is confusing. To clarify, we have modified the text (lines 286–292) and Figure 7.

=====

Reviewer #4 (Remarks to the Author):

In this paper, the authors determined the outward-facing transient state of human ATP-binding cassette transporter 6 of W546A mutant using cryo-EM. Previously, the occluded structures and inward structures were determined using cryo-EM and X-ray structures. This is the first study to determine the outward-facing structure thanks to the W546A mutant, where W546 stabilizes the occluded structures significantly.

The structural results have high impacts and the paper including figures are well written. The proposed mechanisms on the role of W546 are likely and have general importance comparing to other ABC transporters as shown in Figure 6 of the main text.

What I am concerned is the large effect on detergent micelles and nanodiscs in Figure 3. Though the same mutant is used, outward-facing structures are dominant in micelles, while inward-facing structures are dominant in nanodisc. My question is how the latter structures are stabilized without W546. Is it possible to show atomistic structures of the inward-facing form in nanodisc?

>> To address this question, we conducted a thorough re-examination and analysis of the cryo-EM map of the W546A mutant reconstituted in nanodiscs. As a result, we made an intriguing discovery: in the nanodisc environment, a phospholipid molecule is embedded in the hydrophobic groove of the TMD surface of ABCB6 (Figure 4). This phospholipid molecule acts as a latch, effectively stabilizing the occluded conformation of ABCB6, even in the absence of the W546 residue. The revised manuscript

now includes a section to discuss the potential role of membrane lipids in modulating the conformational dynamics of ABCB6. Please see lines 169–197 for a full description.

Related question is that if you change the lipid components in nanodisc, is the structure changed from inward-facing/occluded states to outward-facing structures? I would like to hear more clear explanations about how lipid molecules can replace the interaction from W546, if possible.

>> We have addressed this point above.

Computational works, for instance, molecular modeling and simulation, might be useful for this purpose. I don't request the authors to include such computational works in the paper, since it would take more time and effort before the publication. However, it would be useful to discuss such possibility in Discussion section and to attempt such simulations in the later works.

>> The revised manuscript now includes a section discussing the need for future computational simulation studies to provide clearer explanations of how lipid molecules can compensate for the loss of the W546 stacking interaction. Please see lines 281–285.

Reviewers' comments:

Reviewer #1 (Remarks to the Author):

My comments were addressed with new discussion. However, I still think that the inward facing and the post-occluded structures found in the presence of vanadate are not transport-competent and therefore cannot be used to infer details of the catalytic cycle. Along these lines, the suggested role of W546 in "promoting a shift toward the outward-facing state" is based on irrelevant (i.e. non-functional) structures. The different results in nanodiscs or lipids further suggest that the observed structures are influenced by artefacts related to the crystallization conditions. The main novelty of this study- the outward facing ABCB6 structure- essentially recapitulates known OF ABC structures, without adding any significant new information. Molecular details of the suggested transport pathway remain speculative, without any experimental support.

Reviewer #2 (Remarks to the Author):

In this revised version, the authors have corrected the catalytic cycle (Fig 6) which is now in agreement with many biochemical data, including their new structures, and they added new data that further increase the quality of the manuscript and support their conclusions.

In the text, though, one sentence is still a bit confusing : line 252, « ATP hydrolysis followed by subsequent release of ADP and Pi elicits conformational relaxation, resetting the transporter in the apo state in preparation for another catalytic cycles ». I suggest to start directly the sentence by «Subsequent release of ADP and Pi elicits conformational relaxation, resetting the transporter in the apo state in preparation for another catalytic cycles ».

In addition, when it is written « This preference can be attributed, at least in part, to the high thermodynamic threshold between the inward-facing and outward-facing states under substrate-free conditions, as evidenced by the low ATPase activity of hABCB6core » and also regarding the effect of the Trp mutation with the higher basal ATPase activity than the WT, the authors might quote the article by Orelle et al. (Trends Microbiol. 2023 Mar;31(3):233-241) that strongly supports their conclusions.

Jean-Michel Jault

Reviewer #3 (Remarks to the Author):

While this revised revision shows some improvement it still does not address the main concern: the mechanistical interpretation of the results obtained in detergent solution. The detergent environment has been shown on multiple occasions to alter the conformational dynamics of ABC transporters. The manuscript needs to be rewritten to convey this. As it stands, the discussion and take-home message revolve around the detergent results while those obtained in nanodisks are mentioned as an afterthought.

The ATPase results in lipids should be presented and discussed in the main text, as these are the relevant functional studies. These results should provide the base for all subsequent analysis/conclusions. The detergent work is much less relevant.

"Our kinetic values differ somewhat from previously reported values...."

The values should be explicitly given. The reader should not be expected to search for them. In addition, the stability of the transporter has nothing to do with the kinetic values. If the differences

are very big this should be considered and explained.

“As previously discussed, functionally incompetent molecules may contribute to the predominance of the inward-facing state since they remain recalcitrant to transitioning to the outward-facing conformation. ”

The authors continue to interpret the results as if the detergent environment and the mutation is more mechanistically relevant than the WT in nanodisks. I find this difficult to understand and unacceptable.

From the PBP letter:

“However, a single mutation of W546 to alanine is sufficient to alter the energy landscape among conformations, making the outward-facing state the most stable and lowest energy (Fig. 6b)”

While this is true in detergent solution, this is not the case in nanodisks, broadly considered to be a better a membrane-mimetic environment. How is this fact reflected in the proposed mechanism? >>
We agree that this sentence is confusing. To clarify, we have modified the text (lines 286–292) and Figure 7.

Lines 286–292 do nothing to clarify this point.

Reviewer #4 (Remarks to the Author):

In the previous review, my biggest concern was why the detergent and nanodisc change the ABC transporter's structure so drastically. In the revised manuscript, they found a POPE lipid molecule in the nanodisc structure and discussed the functional roles. It improved significantly our understanding of the ATP transporter's function and structure relationship.

Comments from Reviewers (Bold) and Author Responses

Reviewer #1 (Remarks to the Author):

My comments were addressed with new discussion. However, I still think that the inward facing and the post-occluded structures found in the presence of vanadate are not transport-competent and therefore cannot be used to infer details of the catalytic cycle. Along these lines, the suggested role of W546 in “promoting a shift toward the outward-facing state” is based on irrelevant (i.e. non-functional) structures.

>> We agree with the Reviewer that caution must be exercised when interpreting the transport ability of ABCB6 using the structures determined in the presence of vanadate. Nevertheless, we would like to draw the Reviewer’s attention to previous studies in which vanadate was employed to study the structure and functions of numerous ABC transporters. It is well-known that vanadate traps the ATP hydrolysis transition state of ABC transporters in a stable outward-facing or post-occluded conformation, which has provided valuable insights into the dynamic events of their transport cycles (*see references below).

>> Additionally, like the W546A mutant, the introduction of mutations that modulate equilibration states or stabilize specific functional conformations is a widely accepted technique for the structural analysis of ABC transporters (**see references below). These mutations enable the capture of static snapshots of transporters in different functional states, aiding the interpretation of their complicated catalytic reactions. Therefore, we respectfully disagree with the Reviewer that the post-occluded structure obtained in the presence of vanadate, as well as the structure obtained with the W546A mutant, lacks functional significance. On the contrary, we would like to stress that these approaches have been validated within the field of ABC transporters and have provided valuable insights into the mechanisms of these transporters.

* Ward et al., PNAS, 2007, 104 (48) 19005-19010

* Hofmann et al., 2019, Nature, 571, 580–583

* Harris et al., Nat Comm, 2021, 12(1):5254

* Oldham et al., PNAS, 2011, 108 (37) 15152-15156

** Fan et al., PNAS, 2020, 117(32):19228-19236

** Matsuoka et al., Protein Sci, 2021, 30(5):1064-1071

The different results in nanodiscs or lipids further suggest that the observed structures are

influenced by artefacts related to the crystallization conditions.

>> We have clarified that the structures were determined using cryo-EM, NOT crystallography.

>> Many ABC transporters have exhibited differences in conformation and ATPase activity between detergent and lipid environments (***) see references below). In light of this observation, we argue that the structural and functional differences of ABCB6 between detergent and nanodiscs should not be interpreted as artifacts of the experimental conditions. Rather, they reflect the inherent complexity and adaptability of ABCB6 responses to its environment. We have further revised the manuscript to address these aspects of the transporter (lines 187-191).

*** Zoghbi et al., J. Biol. Chem., 291 (2016), 4453-4461

*** Ambudkar et al., Methods Enzymol., 292 (1998), 492-504

The main novelty of this study- the outward facing ABCB6 structure- essentially recapitulates known OF ABC structures, without adding any significant new information. Molecular details of the suggested transport pathway remain speculative, without any experimental support.

>> We understand the Reviewer's concern about the novelty of the outward-facing ABCB6 structure. While it is true that this conformation shares similarities with those of other ABC structures, we firmly believe that this study advances our understanding of the unique properties of the ABCB6 conformational cycle that facilitates the export of substrates from cells.

>> Additionally, we respectfully disagree with the Reviewer's view that the molecular details of the proposed transport pathway are speculative. Our analysis of the ABCB6-mediated substrate transport mechanism is firmly grounded in the experimental evidence obtained in this study and previous findings. Nonetheless, in response to the Reviewer's concerns, we have revised the manuscript to emphasize the importance of complementing the current findings with further experimental and computational data. Please see lines 280–295.

Reviewer #2 (Remarks to the Author):

In this revised version, the authors have corrected the catalytic cycle (Fig 6) which is now in agreement with many biochemical data, including their new structures, and they added new data that further increase the quality of the manuscript and support their conclusions.

In the text, though, one sentence is still a bit confusing : line 252, « ATP hydrolysis followed by

subsequent release of ADP and Pi elicits conformational relaxation, resetting the transporter in the apo state in preparation for another catalytic cycles ». I suggest to start directly the sentence by «Subsequent release of ADP and Pi elicits conformational relaxation, resetting the transporter in the apo state in preparation for another catalytic cycles ».

>> We have revised the text as suggested.

In addition, when it is written « This preference can be attributed, at least in part, to the high thermodynamic threshold between the inward-facing and outward-facing states under substrate-free conditions, as evidenced by the low ATPase activity of hABCB6core » and also regarding the effect of the Trp mutation with the higher basal ATPase activity than the WT, the authors might quote the article by Orelle et al. (Trends Microbiol. 2023 Mar;31(3):233-241) that strongly supports their conclusions.

>> We have cited the reference as suggested.

Reviewer #3 (Remarks to the Author):

While this revised revision shows some improvement it still does not address the main concern: the mechanistical interpretation of the results obtained in detergent solution. The detergent environment has been shown on multiple occasions to alter the conformational dynamics of ABC transporters. The manuscript needs to be rewritten to convey this. As it stands, the discussion and take-home message revolve around the detergent results while those obtained in nanodisks are mentioned as an afterthought. The ATPase results in lipids should be presented and discussed in the main text, as these are the relevant functional studies. These results should provide the base for all subsequent analysis/conclusions. The detergent work is much less relevant.

>> In the revised version, we now mention and discuss the results of measurements of the ATPase activity of ABCB6 in nanodisks in the main text (lines 143–148).

“Our kinetic values differ somewhat from previously reported values....”

The values should be explicitly given. The reader should not be expected to search for them. In addition, the stability of the transporter has nothing to do with the kinetic values. If the differences are very big this should be considered and explained.

>> As suggested, the manuscript has been modified to include the results of the previous enzymatic activity analysis of ABCB6. We have also removed the sentence claiming that the kinetic values are

linked to protein stability. Please see lines 141–158.

“As previously discussed, functionally incompetent molecules may contribute to the predominance of the inward-facing state since they remain recalcitrant to transitioning to the outward-facing conformation. “The authors continue to interpret the results as if the detergent environment and the mutation is more mechanistically relevant than the WT in nanodisks. I find this difficult to understand and unacceptable.

>> We deleted this sentence from the revised manuscript.

From the PBP letter:

“However, a single mutation of W546 to alanine is sufficient to alter the energy landscape among conformations, making the outward-facing state the most stable and lowest energy (Fig. 6b)”

While this is true in detergent solution, this is not the case in nanodisks, broadly considered to be a better a membrane-mimetic environment. How is this fact reflected in the proposed mechanism?

>> We agree that this sentence is confusing. To clarify, we have modified the text (lines 286–292) and Figure 7.

Lines 286–292 do nothing to clarify this point.

>> We have revised the text (lines 299–310)

Reviewer #4 (Remarks to the Author):

In the previous review, my biggest concern was why the detergent and nanodisc change the ABC transporter's structure so drastically. In the revised manuscript, they found a POPE lipid molecule in the nanodisc structure and discussed the functional roles. It improved significantly our understanding of the ATP transporter's function and structure relationship.

>> Thank you for your thoughtful comments.